# Neuc-MDS: Non-Euclidean Multidimensional Scaling Through Bilinear Forms

Chengyuan Deng, Jie Gao, Kevin Lu, Feng Luo, Hongbin Sun, Cheng Xin*

## Abstract

We introduce **N**on-**Euc**lidean-**MDS** (Neuc-MDS), an extension of classical Multidimensional Scaling (MDS) that accommodates non-Euclidean and non-metric inputs. The main idea is to generalize the standard inner product to symmetric bilinear forms to utilize the negative eigenvalues of dissimilarity Gram matrices. Neuc-MDS efficiently optimizes the choice of (both positive and negative) eigenvalues of the dissimilarity Gram matrix to reduce STRESS, the sum of squared pairwise error. We provide an in-depth error analysis and proofs of the optimality in minimizing lower bounds of STRESS. We demonstrate Neuc-MDS's ability to address limitations of classical MDS raised by prior research, and test it on various synthetic and real-world datasets in comparison with both linear and non-linear dimension reduction methods.

## 1 Introduction

Many datasets in applications adopt dissimilarities that are non-Euclidean and/or non-metric. Examples of such popular dissimilarity measures [19, 7] include Minkowski distance $(L_p)$, cosine similarity, Hamming, Jaccard, Mahalanobis, Chebyshev, and KL-divergence. Studies in psychology have long recognized that human perception of similarity is not a metric [42]. Further, dissimilarity matrices with negative entries (e.g.: cosine similarity, correlation, signed distance) have also been widely used in various problems. Negative inner product norms also have deep connections to hyperbolic spaces as well as the study of spacetime in special relativity theory.

In many machine learning practices, embedding in low dimensional vector space is often explicitly or implicitly done within the data processing pipeline. Such embedding may already need to consider more general dissimilarities. For example, one branch of graph learning adopts embedding in non-Euclidean spaces (e.g., hyperbolic spaces [9]), and machine learning for physics model and data (AI4Sicence) needs to consider more general inner product norms [22]. In transformer models [45, 13], the attention mechanism can also be viewed as learning a general bilinear form on tokens. Despite the wide adoption of general dissimilarity measures in practice, theoretical study of embedding and dimension reduction for non-Euclidean non-metric data appears to be still very limited.

In this paper we consider one of the most classical algorithms for data embedding and dimension reduction — multidimensional scaling – and develop a non-Euclidean, non-metric version with theoretical performance guarantee.

**Background on MDS.** Classical multidimensional scaling (cMDS) takes as input a *Euclidean distance matrix (EDM)*, i.e., a symmetric matrix $D \in \mathbb{R}^{n \times n}$ where each entry is the squared Euclidean distance between two points in Euclidean space, and recovers the Euclidean coordinates. Using a standard double centering trick, $D$ can be turned into a

---

*Rutgers University. {cd751,jg1555,kll160,fluo,hongbin.sun,cx122}@rutgers.edu

38th Conference on Neural Information Processing Systems (NeurIPS 2024).

Gram matrix $B = X^T X$ where $X$ encodes the Euclidean coordinates. For the purpose of producing a low-dimensional vector, classical MDS takes $k$ eigenvectors corresponding to top $k$ largest eigenvalues of the Gram matrix $B$. This minimizes the *strain*, the difference (Frobenius norm) in terms of the Gram matrix.

When the input distance matrix is not a Euclidean distance matrix, this problem is called metric MDS [1]. Metric MDS considers the minimization of *STRESS* [28], defined as the sum of squared difference of pairwise embedding distances to the input dissimilarities. Minimizing STRESS makes the problem to be non-linear and there is no closed-form solution, though one can use either gradient descent or Newton's method [35, 26]. Nevertheless, in practice, cMDS is often applied for non-Euclidean distance matrix. In this case, the centered matrix $B$ is no longer positive semi-definite. The common practice is keep the top positive eigenvalues and throw away the negative eigenvalues.

Two recent papers [41, 38] pointed out that classical MDS produces suboptimal solutions on non-Euclidean distance matrix when considering STRESS. This is not a surprise, since cMDS, minimizing strain, does not minimize STRESS. However, a more problematic issue is that when using more dimensions in cMDS (i.e., increasing $k$), the STRESS error first drops and then increases. We call this phenomena *Dimensionality Paradox*. It is theoretically unsatisfactory and counter-intuitive that embeddings by classical MDS using more dimensions could yield worse results.

The error analysis in [38] sheds light on this issue. When the input matrix is Non-Euclidean, the negative eigenvalues carry crucial information. cMDS, keeping only positive eigenvalues, is intrinsically biased – the more positive eigenvalues used the more it deviates from the input data. A real eradication of this issue must address the root cause, i.e., applying an algorithm meant for Euclidean geometry on non-Euclidean data.

**Our Contributions.** We extend multidimensional scaling to non-Euclidean geometry, by generalizing the dissimilarity function from the standard inner product (which defines Euclidean geometry) to the broader family of symmetric bilinear forms $\Phi(u, v) = u^T A v$, where the symmetric matrix $A$ does not have to be positive semi-definite. For dimension reduction, we look for both a low dimensional vector representation and an associated bilinear form, that together approximate the input dissimilarity matrix with minimum STRESS error. Specifically, the key contributions are as follows:

- We conduct an in-depth analysis on STRESS error for any chosen subset of $k < n$ eigenvalues of the input centered dissimilarity matrix. We propose Neuc-MDS, an efficient algorithm that finds the best subset of eigenvalues to minimize a lower bound of STRESS.

- Beyond the constraints of eigenvalue subsets, we extend our findings to the general linear combinations of eigenvalues. Our advanced algorithm, Neuc-MDS$^+$, finds the best linear combination of eigenvalues to minimize the lower bound objective.

- We provide theoretical analysis for the asymptotic behavior of cMDS and Neuc-MDS on random symmetric matrices. First, both necessarily produce large STRESS if the target dimension $k = o(n)$ – on completely unstructured data aggressive dimension reduction shall not be expected. Further, when $k = \Theta(n)$, the STRESS of Neuc-MDS monotonically decreases to 0 while the STRESS of cMDS increases and eventually reaches a plateau.

- Empirically we evaluate Neuc-MDS and Neuc-MDS$^+$ on ten diverse datasets encompassing different domains. The experiment results show that both methods substantially outperform previous baselines on STRESS and average distortion, and fully resolve the issue of *dimensionality paradox* in cMDS. Our codes are available on Github[2].

## 2   Related Work

Our work is in the general family of similarity learning [25, 3] with dimension reduction, going beyond metric learning and embedding. Due to the huge amount of literature on this topic we only mention those that are most relevant.

---

[2] https://github.com/KLu9812/MDSPlus

**MDS Family and Embedding in Euclidean Spaces.** As one of the most useful embedding and dimension reduction techniques in practice, the MDS family has many variants. Non-metric MDS [36, 37] considers a monotonically increasing function $f$ on input dissimilarity and minimizes STRESS between $\{f(D_{ij})\}$ and embedded squared Euclidean distances. Generalized multidimensional scaling (GMD) considers the target space as an arbitrary smooth surface [6]. In addition, non-linear dimension reduction methods such as Isomap [39], Laplacian Eigenmaps [2], LLE [34], t-SNE [23, 44] consider data points from a non-linear high-dimensional manifold and extract distances defined *on* the manifold. In all these methods, the points are still embedded in Euclidean spaces. Some of these methods such as Isomap directly apply cMDS as the final step. If the dissimilarity is highly non-Euclidean, we can replace cMDS by Neuc-MDS to get performance improvement.

**Dimension Reduction in Non-Euclidean Spaces.** There is also prior work that finds embedding of manifold data in non-Euclidean spaces, e.g., on a piece-wise connected manifold [50, 5], on a sphere [11, 15, 48], and in hyperbolic spaces [15, 48]. Very recently, there is study of Johnson–Lindenstrauss style dimension reduction for weighted Euclidean space [30], hyperbolic space [4], as well as PCA [8], dimension-reduction [16], and t-SNE in hyperbolic space [20]. Our dissimilarity function generalizes beyond hyperbolic distances.

## 3 Dimension Reduction with Bilinear Forms

Let $P$ denote a dataset of size $n$. Let $D \in \mathbb{R}^{n \times n}$ be the dissimilarity matrix of dataset $P$, $D_{ij} = D_{ji}$ is a real-valued symmetric dissimilarity measure between pair $p_i, p_j \in P$, $i \neq j \in [n]$, and all diagonal entries $D_{ii} = 0$ (i.e., $D$ is a hollow matrix). $D$ is the analog of squared Euclidean distance matrix in classical MDS. But here $D$ is not necessarily Euclidean, may not be a metric (e.g. violating triangle inequality) and may have negative entries. Our goal is to obtain (1) a low-dimensional vector representation for each element in $P$, and (2) a function $f$ that computes a dissimilarity measure using the calculated low dimensional vectors. Since the input dissimilarities are not necessarily Euclidean nor a metric, we look for the function $f$ beyond Euclidean distances, but stay within a broader family of inner products or bilinear forms.

A bilinear form $\Phi$ on a vector space $V$ is a function $\Phi : V \times V \to \mathbb{R}$ which is linear in each variable when the other variable is fixed. More precisely, $\Phi(au + v, w) = a\Phi(u, w) + \Phi(v, w)$ and $\Phi(w, au + v) = a\Phi(w, u) + \Phi(w, v)$ for all $u, v, w \in V$ and any scalar $a$. We only consider symmetric bilinear forms $\Phi$, i.e., $\Phi(u, v) = \Phi(v, u)$. A bilinear form is positive definite (or positive semi-definite) if $\Phi(u, u) > 0$ for $\forall u \neq 0$ (or $\Phi(u, u) \geq 0$). Symmetric matrices and symmetric bilinear forms are two sides of a coin. Namely, fix a basis $W = \{w_1, ..., w_n\}$ of the vector space, there is a one-to-one correspondence between them. That is, $A = [\Phi(w_i, w_j)]_{n \times n}$ is a symmetric matrix. Conversely, give a symmetric matrix $A$, one defines a symmetric bilinear form $\Phi(u, v) = u_W^T A v_W$ where $u_W$ is the coordinate of vector $u$ in the basis $W$.

Formally, we have the following problem definition.

**Definition 1** (Non-Euclidean Dimension Reduction). *Given a symmetric dissimilarity matrix $D$ of a dataset $P$ of size $n$ and a natural number $k \leq n$, find a collection of $n$ $k$-dimensional vectors $\hat{P} = (\hat{p}_1, \cdots, \hat{p}_n : \hat{p}_i \in \mathbb{R}^k)$ with a bilinear form $f : \mathbb{R}^k \times \mathbb{R}^k \to \mathbb{R}$, $f(u, v) = u^T A v$, such that the STRESS error $||\hat{D} - D||_F^2$ for the dissimilarity matrix $\hat{D}$ of $\hat{P}$ given by $\hat{D}_{ij} = f(\hat{p}_i, \hat{p}_j)$ is minimized.*

### 3.1 MDS in the Lens of Bilinear Forms

A special case of a symmetric bilinear form is the standard inner product $\langle, \rangle$ on $\mathbb{R}^n$. Indeed the inner product $\langle u, v \rangle = u^T v$ is a symmetric positive definite bilinear form. The inner product of $u - v$ and $u - v$ is precisely the *squared* Euclidean distance $||u - v||^2$. The Euclidean space is $\mathbb{R}^n$ equipped with the standard inner product. Thus metric geometry of Euclidean space is governed by the inner product. On the other hand, a basic theorem of linear algebra states that a finite dimensional vector space equipped with a symmetric *positive definite* bilinear form is *isometric* to the Euclidean space of the same dimension.

Thus geometry of a positive definite symmetric bilinear form, or symmetric positive definite matrices, is nothing but Euclidean geometry.

When $D$ represents the inner products of pairwise differences (i.e., squared Euclidean distances) of $n$ points $P$ in $\mathbb{R}^d$, $D$ is called a *Euclidean distance matrix (EDM)*. It can be shown that by taking $B = -\frac{1}{2}CDC$, where $C = I - \frac{1}{n}\mathbf{1}_n\mathbf{1}_n^T$ is the centering matrix and $\mathbf{1}_n$ is a vector of ones, one obtains the Gram matrix $B = X^TX$ where $X$ is $d \times n$ dimensional matrix of the $n$ coordinates of dimension $d$. The matrix $B$ is a symmetric positive semi-definite matrix. Thus using eigendecomposition of $B$ one can recover the coordinates $X$. This procedure is *classical multidimensional scaling (cMDS)* [40]. Furthermore, if one would like to use $k$-dimension coordinates with $k < d$, classical MDS suggests to take the eigenvectors corresponding to the $k$ largest positive eigenvalues of the Gram matrix $B$. This minimizes the *strain*, the difference (Frobenius norm $\|\cdot\|_F$) in terms of the Gram matrix.

$$X_{cmds} \triangleq \arg\min_{X \in \mathbb{R}^{n \times k}} \|X^TX - (\frac{-CDC}{2})\|_F^2. \tag{1}$$

Now consider a general symmetric dissimilarity matrix $D$ of size $n \times n$, it naturally associates $\mathbb{R}^n$ with a symmetric bilinear form $\Phi$. In many real-world situations, the dissimilarity matrix is not positive, nor negative definite, i.e., the bilinear form is indefinite. This means the intrinsic geometry $(\mathbb{R}^n, \Phi)$ is non-Euclidean. The relationship between the Gram matrix and square distance matrix still holds for indefinite bilinear forms. See Appendix A

In practice, classical MDS is often the default choice even when the input dissimilarity matrix $D$ is not an EDM, i.e., the centered matrix $B = -\frac{1}{2}CDC$ is not positive semi-definite. Classical MDS, which simply drops negative eigenvalues of $B$ to produce positive semi-definiteness does not respect the geometry well. Indeed, multiple researchers have observed suboptimal and counter-intuitive performance [41, 38]. For example, increasing $k$ may lead to increased STRESS error – keeping more dimensions makes the approximation worse! On a second thought, such results are not surprising. If the input data does not carry Euclidean geometry, forcing it through a procedure for Euclidean geometry is fundamentally problematic.

Our main observation is that a vector space with an indefinite symmetric bilinear form has its own intrinsic geometry. This geometry, even though may not be metrical, carries the most accurate information about the dissimilarity matrix and the datasets. Suppose the centered matrix $B = -\frac{1}{2}CDC$ has $p > 0$ positive eigenvalues and $q > 0$ negative eigenvalues. The associated indefinite bilinear form $\Phi$ has signature $(p, q)$. One such example is

$$\Phi(u, v) = \sum_{i=1}^{p} u_i v_i - \sum_{i=p+1}^{p+q} u_i v_i. \tag{2}$$

The geometry of a finite dimensional vector space with a symmetric bilinear form of signature $(p, q)$ where $p, q > 0$ is much less developed compared to Euclidean geometry. When $q = 1$, this is the Minkowski geometry and is closely related to relativity theory in physics and hyperbolic geometry [33] in mathematics. For general $(p, q)$, one should probably abandon the notion of distance for indefinite spaces. Namely, the expression $\Phi(u - v, u - v)$, even if it is positive, should not be considered as the square of the distance between two points $u, v$. According to [33], one calls $\sqrt{\Phi(u - v, u - v)}$ the Lorenzian distance between $u, v$. Despite the term distance in the name, the Lorenzian distance does not satisfy triangle inequality in general.

## 3.2   Non-Euclidean MDS

We propose *Non-Euclidean MDS*, a novel linear dimension reduction technique using bilinear forms. For a vector $v$, we use $\text{Diag}(v)$ as the diagonal matrix with $v$ along the main diagonal and zero everywhere else. For any given symmetric dissimilarity matrix $D \in \mathbb{R}^{n \times n}$, let the eigen decomposition of the centralization of $D$ be given as follows:

$$-CDC/2 = U\Lambda U^T \tag{3}$$

where $U \in \mathbb{R}^{n \times n}$ is the orthogonal matrix of eigenspace and $\Lambda \in \mathbb{R}^{n \times n}$ is the diagonal matrix $\Lambda =: \mathrm{Diag}(\boldsymbol{\lambda})$ with eigenvalues $\boldsymbol{\lambda} \triangleq (\lambda_1, \cdots, \lambda_n)^T$, $\lambda_1 \geq \lambda_2 \geq \cdots \geq \lambda_n$. By using an algorithm that will be discussed in Section 4 we will choose $k$ of the eigenvalues, represented by a binary indicator vector $\boldsymbol{w} = (w_1, \cdots, w_n) \in \{0,1\}^n$ with a value of 1 (or 0) indicating the corresponding eigenvalue is chosen (or not chosen). Let

$$X = \sqrt{\Lambda} \cdot \mathrm{Diag}(\boldsymbol{w}) \cdot U^T =: (X_1, \cdots, X_n), \qquad (4)$$

Note that $\sqrt{\Lambda}$ contains complex numbers for non-PSD matrix $D$. Since $\boldsymbol{w}$ has only $k$ non-zero values, we can drop the $n - k$ zero rows in $X$ (corresponding to the eigenvalues not selected) and have a $k$-dimensional vector representation of the data. Now we can derive dissimilarities by defining

$$\hat{D}_{i,j} \triangleq (X_i - X_j)^T(X_i - X_j), \quad \hat{D} = \mathrm{DIS}(X) := (\hat{D}_{i,j}).$$

See 1 for details.

---

**Algorithm 1**: Non-Euclidean Multidimensional Scaling

---

**Input**: $n \times n$ dissimilarity matrix $D$, integer $k \leq n$.
**Output**: $k \times n$ matrix $X$ of $k$-dim vectors
$B = -\frac{1}{2} CDC$, where $C = I - \frac{1}{n} \mathbf{1}_n \mathbf{1}_n^T$ ;
Compute eigenvalue vector $\boldsymbol{\lambda}$ and eigenvectors $U$ of $B$: $\boldsymbol{\lambda} = (\lambda_1, \lambda_2, \cdots \lambda_n)^T$ with
$\lambda_1 \geq \lambda_2 \geq \cdots \geq \lambda_n$;
Compute the indicator vector of $k$ selected eigenvalues $\boldsymbol{w} = \text{EV-Selection}(\boldsymbol{\lambda}, k)$;
Compute $X$ by $\sqrt{\Lambda} \cdot \mathrm{Diag}(\boldsymbol{w}) \cdot U^T$ with $n - k$ zero rows dropped;

---

In the description above, we allow the coordinates in $X$ to take complex numbers, such that we can take $\hat{D}_{i,j}$ to be the standard dot product of $X_i - X_j$ with $X_i - X_j$. Alternatively, we can keep $X$ to take real coordinates, i.e., $X = \sqrt{\Lambda'} \cdot \mathrm{Diag}(\boldsymbol{w}) \cdot U^T$ with $\Lambda' = \mathrm{Diag}(|\lambda_1|, \cdots, |\lambda_n|)$. Again we drop the $n - k$ zero rows in $X$ and take a bilinear form $f(u, v) = u^T A v$, where $A$ is a $k$ by $k$ diagonal matrix, with the element at $(i, i)$ to be 1 (or $-1$) if the corresponding eigenvalue chosen is positive/negative. Notice that the bilinear form takes precisely the format of $(p, q)$-distance as in Equation (2).

We have a few remarks in place: First we are not throwing away the negative eigenvalues. As will be explained later we actually keep eigenvalues of largest magnitude and some could be negative. As a consequence $\hat{D}$ may have negative real values, which is expected as we are moving away from Euclidean distances and input entries in $D$ may even start to be negative. Second, when $D$ is an EDM, i.e., all eigenvalues are non-negative, Neuc-MDS reduces to classical MDS. Last, similar to cMDS, our method also starts with computing the eigenvalues of the Gram matrix. For large datasets, fast (approximation) algorithm for partial SVD can also be applied to our methods. For example, on $n \times n$ symmetric matrices, using power methods one can iteratively compute partial SVD up to $k$ largest/smallest eigenvalues in $O(kn^2)$. With randomness introduced, it can be reduced to $O(\log k \cdot n^2)$ [18, 27]. For really large datasets, the dissimilarity matrix with size $O(n^2)$ might already be too large to be acquired or stored, one may extend MDS through local embedding methods like Landmark MDS [12] or Local MDS [10]. This approach can also be applied to neuc-MDS to substantially speed up computation without suffering too much on performance (See Section 6 for empirical results).

## 4 Theoretical Results for Non-Euclidean MDS

To establish the foundation of theoretical analysis, we first analyze the STRESS error of Neuc-MDS and decompose it into three terms. Next we show an efficient algorithm that minimizes the first two terms that dominate. Last, we examine Neuc-MDS and classical MDS on random Gaussian matrices.

## 4.1 Error Analysis

Inspired by the analysis of STRESS of classical MDS [38], we adopt a similar approach and decompose the STRESS error into three terms. Let $\boldsymbol{\lambda} \in \mathbb{R}^n$ be the vector of all eigenvalues $\boldsymbol{\lambda} = (\lambda_1, \cdots, \lambda_n)^T$, $\boldsymbol{\lambda}^{(2)} \in \mathbb{R}^n$ be the vector of all squared eigenvalues $\boldsymbol{\lambda}^{(2)} = (\lambda_1^2, \cdots, \lambda_n^2)^T$, $\bar{\boldsymbol{w}} \triangleq \mathbf{1}_n - \boldsymbol{w}$ be the indicator vector of dropped eigenvalues, $\bar{\boldsymbol{w}} \in \{0,1\}^n$, and $\odot$ be the Hadamard product. We have the following result with proof in Appendix B.

**Theorem 2.** *It holds that* $\|\hat{D} - D\|_F^2 = C_1 + C_2 + C_3$*, where*

$$C_1 = 4\bar{\boldsymbol{w}}^T \boldsymbol{\lambda}^{(2)}, \; C_2 = 4(\bar{\boldsymbol{w}}^T \boldsymbol{\lambda})^2, \; C_3 = 2n\|(U \odot U)(\bar{\boldsymbol{w}} \odot \boldsymbol{\lambda})\|_F^2 - C_2/2.$$

Note that $C_1/4$ is the *sum of squared* eigenvalues that are dropped, and $C_2/4$ is the *square of the sum* of eigenvalues that are dropped. Individually, we can minimize $C_1$ by keeping eigenvalues of large absolute value; and $C_2$ by balancing the dropped eigenvalues such that the summation has a small magnitude. For term $C_3$, from Equation 21 in Proof B, we know $C_3 \geq 0$. In [38] it is argued that if one takes the approximation $\|(U \odot U)(\bar{\boldsymbol{w}} \odot \boldsymbol{\lambda})\|_F^2 \approx \|\frac{1}{\sqrt{n}}\mathbf{1}_n\mathbf{1}_n^T(\bar{\boldsymbol{w}} \odot \boldsymbol{\lambda})\|_F^2$ for a random orthogonal matrix $U$, then $C_3 \approx 0$. Although it is empirically observed in [38] that $C_3$ is roughly constant and hence negligible for optimization, there are some cases in our experiments in which $C_3$ is not negligible.

In light of Theorem 2, we would like to approximately optimizing the STRESS by minimizing the lower bound $C_1 + C_2$, which can be formulated as a quadratic integer programming problem: Given a set of $n$ values $\mathcal{L} = \{\lambda_i\}$ and a positive integer $k > 0$, choose a $k$-subset $S \subseteq \mathcal{L}$ such that

$$\min_{S \subseteq \mathcal{L}, |S| = k} \sum_{\lambda \in \mathcal{L} \setminus S} \lambda^2 + \Big( \sum_{\lambda \in \mathcal{L} \setminus S} \lambda \Big)^2. \tag{5}$$

When $\lambda_i$'s are all positive, the best choice is to take the $k$ largest eigenvalues of $\mathcal{L}$, i.e., the cMDS' solution. However, with a mixture of positive and negative eigenvalues, taking the top $k$ largest eigenvalues is no longer optimal — specifically, as $k$ increases, the first error term is monotonically reduced but the second error term could start going up. In the following subsection, we discuss an optimal algorithm to solve Equation (5).

## 4.2 An Optimal Algorithm for Eigenvalue Selection

The optimization problem described in Equation (5) is a special case of the family of quadratic integer programming problems. Though in general, quadratic integer programming is NP-hard [31], we show that this particular one is actually solvable in polynomial time. Formally, we have:

**Theorem 3.** *For the optimization problem defined in (5), there exits an optimal solution with $r$ largest positive values and $s$ smallest negative values in $\mathcal{L}$, $r + s = k$. And, there is an $O(n)$-algorithm that outputs an optimal solution.*

Our algorithm (2) is greedy and iteratively selects the eigenvalue with the highest absolute value. Specifically, let $S$ be the set of selected eigenvalues and $H(S)$ be the sum of eigenvalues not selected. Initially $S = \emptyset$. In each iteration, if $H(S) < 0$, select the negative eigenvalues remained of the greatest magnitude and add it to $S$; if $H(S) > 0$, pick the largest positive one. If $H(S) = 0$, pick the eigenvalue with the greatest magnitude. The complete proof of Theorem 3 is delayed to Appendix C.

## 4.3 Analysis on Random Symmetric Matrices

Here, we analyze the important error term $C_1 + C_2$ in Theorem 2 for cMDS and Neuc-MDS when the input (centered) dissimilarity matrix is a symmetric random matrix. Our analysis is established upon Wigner's famous Semicircle Law [46, 47], which states the following.

**Proposition 4** (Semicircle Law [47])**.** *Suppose $B \in \mathbb{R}^{n \times n}$ is a symmetric random matrix where every element is independently distributed with equal densities and second moments $\sigma^2$. Let $S_{a,b}(B)$ be the number of eigenvalues of $B$ that lie in the interval $(a\sqrt{n}, b\sqrt{n})$. Then the expected value $E(S_{a,b}(B))$ of $S_{a,b}(B)$ satisfies*

$$\lim_{n \to \infty} \frac{E(S_{a,b}(B))}{n} = \frac{1}{2\pi\sigma^2} \int_a^b \sqrt{4\sigma^2 - x^2}\,dx. \tag{6}$$

---

**Algorithm 2**: Eigenvalues Selection for Neuc-MDS

---

**Input**: $\mathcal{L}$: sorted set of $n$ eigenvalues, integer $k$.
**Output**: $k$ eigenvalues $S$
**Initialize** $S = \emptyset$, **repeat**
    Compute $H(S) = \sum_{\lambda \in \mathcal{L} \setminus S} \lambda$
    Select an eigenvalue $\lambda^*$ such that $\lambda^* \cdot H(S) \geq 0$ and $\lambda^*$ has the largest absolute value.
    Add $\lambda^*$ to $S$.
**until** $|S| = k$;

---

Our results are formally stated as follows with proof in Appendix D.

**Theorem 5.** *Suppose a random symmetric matrix $B \in \mathbb{R}^{n \times n}$ where $B_{ij}$ is independently distributed with equal densities and second moments $\sigma^2$, is taken as the centered matrix[3] to classical MDS and Neuc-MDS, both selecting $k$ eigenvalues with $k \leq n$. Let $e_C$ denote the $C_1 + C_2$ error for cMDS and $e_N$ for Neuc-MDS, we have:*

1. *when $k = o(n)$, $e_C \approx n^2 \sigma^2 (1 + \frac{4k^2}{n} - \frac{4k}{n})$, and $e_N \approx n^2 \sigma^2 (1 - \frac{4k}{n})$*
2. *when $k = cn$, with $c \to 1$, $e_N \approx 0$. When $c \geq 1/2$, $e_C \approx 0.1801 \cdot n^3 \sigma^2$.*

By Theorem 5, Neuc-MDS has a strictly better $C_1 + C_2$ error than cMDS. Second, if we take $|S| = k \ll n = |\mathcal{L}|$, the term $C_1 = \sum_{\lambda \in \mathcal{L} \setminus S} \lambda^2$ is almost equal to $\sum_{\lambda \in \mathcal{L}} \lambda^2 \approx n^2 \sigma^2$. Therefore the STRESS error of both classical MDS and Neuc-MDS cannot be very small if the target dimension $k$ is $o(n)$. Lastly, when $k = cn$, with $c \to 1$, $e_N$ is monotonically decreasing and eventually reaches 0. On the other hand, for cMDS, when $c \geq 1/2$, $e_C$ reaches a plateau at about $0.1801 \cdot n^3 \sigma^2$. Notice that this error has an extra factor of $n$ compared to the error for small $k$.

For real world data the Gram matrix is likely far from a random matrix. The analysis above points out that aggressive dimension reduction can be indeed only a luxury for structured data, even if we use inner products that are not limited to Euclidean distances. Second, any real world data carries some random measurement noise. When the scale of such random noise becomes non-negligible, STRESS error introduced by such noise cannot be small with aggressive dimension reduction. We would recommend practitioners to examine the spectrum of eigenvalues to gain insights on the power or limit in reducing dimensions.

## 5 Beyond Binary Selection of Eigenvalues

Both cMDS and Neuc-MDS choose a subset of $k$ eigenvalues from the input Gram matrix. This can be considered as applying $\mathrm{Diag}(\boldsymbol{w})$ to the eigenvalue vector $\Lambda$ (Equation 4) to filter some eigenvalues out. Such operators can be viewed as a special case of more general low-rank linear maps $T : \mathbb{R}^n \to \mathbb{R}^n$ with $\mathrm{rank}(T) = k \leq n$.

Here we consider a new family of embeddings $\tilde{X} = U\tilde{\Lambda}^{1/2}$ with $\tilde{\Lambda}$ being a rank-$k$ diagonal matrix whose diagonal entries are given by some $\tilde{\boldsymbol{\lambda}} \in \mathbb{R}^n$ with only $k$ nonzero entries. Note that there always exists a rank-$k$ linear operator $T$ such that $\tilde{\boldsymbol{\lambda}} = T(\boldsymbol{\lambda})$. Then we ask if we can improve Neuc-MDS further with a rank-$k$ linear map of $\boldsymbol{\lambda}$? This question can be answered by the following theorem:

**Theorem 6.** *Given a dissimilarity matrix $D \in \mathbb{R}^{n \times n}$ with eigenvalues $\boldsymbol{\lambda} \in \mathbb{R}^n$ of its Gram matrix $-CDC/2 = U\Lambda U^T$, for any rank-$k$ ($k \leq n$) diagonal matrix $\tilde{\Lambda} = \mathrm{Diag}(\tilde{\boldsymbol{\lambda}})$ with $\tilde{\boldsymbol{\lambda}} \in \mathbb{R}^n$ with $k$ nonzero entries only on coordinates given by some index $k$-set $W \subseteq [n]$, let $\boldsymbol{w} := \mathbf{1}_W \in \{0,1\}^n$ be the indicator vector of $W$, and let $\tilde{D}$ be the dissimilarity matrix reconstructed from $\tilde{X} = \tilde{\Lambda}^{1/2} U^T$. Then the STRESS error of $\tilde{D}$ can be expressed as:*

---

[3]Here we take $B$ as the centered matrix. Therefore the diagonal entries of $B$ do not have to be 0. Further, the distribution of centered matrix (ignoring the scaling factor) $CAC$ is the same as the distribution of $\Phi(A)$ plus an additional zero eigenvalue. Therefore, one can consider $\Phi(A)$ sampled from random matrices. When $n \to \infty$, they are asymptotically the same. See more discussion in the appendix (Lemma 12).

$\|\tilde{D} - D\|_F^2 = \tilde{C}_1 + \tilde{C}_2 + \tilde{C}_3$ *with*

$$\tilde{C}_1 := 4\left[\bar{\boldsymbol{w}}^T\boldsymbol{\lambda}^{(2)} + \boldsymbol{w}^T\Delta\boldsymbol{\lambda}^{(2)}\right], \; \tilde{C}_2 := 4\left[\bar{\boldsymbol{w}}^T\boldsymbol{\lambda} + \boldsymbol{w}^T\Delta\boldsymbol{\lambda}\right]^2, \; \tilde{C}_3 := 2n\|(U\odot U)(\Delta\boldsymbol{\lambda})\|_F^2 - \frac{\tilde{C}_2}{2},$$

*where* $\Delta\boldsymbol{\lambda} := \boldsymbol{\lambda} - \tilde{\boldsymbol{\lambda}}$. *The first two terms,* $\tilde{C}_1 + \tilde{C}_2$, *as a lower bound of the STRESS, is minimized as*

$$4\bar{\boldsymbol{w}}^T\boldsymbol{\lambda}^{(2)} + \frac{4(\bar{\boldsymbol{w}}^T\boldsymbol{\lambda})^2}{1+k} \; \text{ with } \tilde{\boldsymbol{\lambda}} \text{ to be } \tilde{\boldsymbol{\lambda}}^* := \boldsymbol{\lambda}\odot\boldsymbol{w} + \frac{\bar{\boldsymbol{w}}^T\boldsymbol{\lambda}}{1+k}\boldsymbol{w}. \tag{7}$$

**Remark 7.** *Recall the decomposition of STRESS in Theorem 2, the lower bound* $4\bar{\boldsymbol{w}}^T\boldsymbol{\lambda}^{(2)} + 4(\bar{\boldsymbol{w}}^T\boldsymbol{\lambda})^2 = C_1 + C_2/(k+1) \leq C_1 + C_2$. *Therefore, it has a better lower bound of STRESS compared to* Neuc-MDS.

Theorem 6 provides a constructive way to obtain the optimal $\tilde{\boldsymbol{\lambda}}$ that (approximately) minimizes the STRESS error for a prefixed $\boldsymbol{w}$ which is determined by an indicator set $W$ served as the constraints of nonzero entries on $\tilde{\boldsymbol{\lambda}}$. The next question is how to find an optimal $k$-set $W$ on which the optimal $\tilde{\boldsymbol{\lambda}}^*$ has the lowest lower bound. Essentially, we need to solve the following optimization problem:

$$\min_{|W|=k}\left[\sum_{i\notin W}\lambda_i^2 + \frac{1}{1+k}(\sum_{i\notin W}\lambda_i)^2\right] \tag{8}$$

**Proposition 8.** *For the optimization problem (8), there exits an optimal solution with* $r$ *largest positive values and* $s$ *smallest negative values in* $\mathcal{L}$, $r + s = k$. *And, there is an* $O(n)$*-algorithm that outputs the optimal solution.*

The algorithm follows a similar greedy manner as EV-SELECTION, with only one adjustment: in each step, compare the two marginal gains provided by the largest positive eigenvalue and the lowest negative eigenvalue among the remained ones, and choose the positive/negative eigenvalue of largest magnitude if the corresponding gain is smaller. We name the algorithm Neuc-MDS$^+$, and present the complete algorithm with its correctness proof in Appendix C.

## 6 Experiments

This section presents experimental results of Neuc-MDS and Neuc-MDS$^+$. First, we evaluate the performance on dissimilarity error of two proposed algorithms comparing with closely-related baselines on three metrics: STRESS, distortion and additive error. Then we show that the *dimensionality paradox* issue observed on cMDS is fully resolved by Neuc-MDS and Neuc-MDS$^+$.

**Synthetic Data.** We introduce two synthetic datasets: *Random-simplex* and *Euclidean-ball*, both with non-Euclidean dissimilarities. See details in Appendix E.1. On a high level, suppose the dataset has size $n$, we construct a *Random-simplex* such that for each vertex, the first $n-1$ coordinates virtually form a simplex, while the last coordinate almost dominates the distances between the other points, which creates a large negative eigenvalue for the Gram matrix. The *Euclidean-ball* dataset (similar to Delft's balls [14]) considers $n$ balls of different radii with the distance of two balls defined as the smallest distance of two points on the two respective balls. The dissimilarities by this definition no longer satisfy triangle inequality.

**Real-world Data.** We consider both genomics and image data. For genomics data, we include 5 datasets from the Curated Microarray Database (CuMiDa) [17], each indicating a certain type of cancer. Following the practice mentioned in [43], pairwise dissimilarities are generated with entropic affinity with the diagonal as zero. We also test three celebrated image datasets: MNIST, Fashion-MNIST and CIFAR-10. The dissimilarity matrix for each dataset captures 1000 images randomly sampled for each class. We use three measures in [38] to calculate the dissimilarities.

An essential guidance of our choice of datasets is the existence of substantial negative eigenvalues. Otherwise, Neuc-MDS and cMDS are equivalent. Figure 1 illustrates the eigenvalue distribution of the Renal dataset. Table 1 shows the basic statistics of each dataset and the number of negative eigenvalues of the Gram matrix. All datasets are non-Euclidean and non-metric.

Figure 1: Negative (red), positive (blue).

Eigenvalues Distribution of Renal

Table 1: Datasets used in experiments.

| Dataset | Size | # $\{\lambda < 0\}$ | Classes | Metric |
|---|---|---|---|---|
| Simplex | 1000 | 900 | N.A. | ✗ |
| Ball | 1000 | 887 | N.A. | ✗ |
| Brain | 130 | 53 | 5 | ✗ |
| Breast | 151 | 59 | 6 | ✗ |
| Colorectal | 194 | 78 | 2 | ✗ |
| Leukemia | 281 | 117 | 7 | ✗ |
| Renal | 143 | 57 | 2 | ✗ |
| MNIST | 1000 | 454 | 10 | ✓ |
| Fashion | 1000 | 429 | 10 | ✓ |
| CIFAR-10 | 1000 | 399 | 10 | ✓ |

**Baselines.** We include cMDS, Lower-MDS [38] and SMACOF (Scaling by MAjorizing a COmplicated Function) [29] as baselines. Lower-MDS [38] looks for a symmetric, low-rank, trace-zero, and positive semi-definite matrix. SMACOF minimizes STRESS using majorization and is one of the best nonlinear optimization algorithms for MDS. All methods are deterministic therefore variance is not concerned.

Table 2: Evaluation Results on STRESS.

| Dataset | cMDS | Lower-MDS | Neuc-MDS | Neuc-MDS$^+$ | SMACOF |
|---|---|---|---|---|---|
| Random-Simplex | 80.520 | 31.542 | 1.179 | **0.194** | 15.962 |
| Euclidean Ball | 36.975 | 17.303 | **1.196** | 1.351 | 4e6 |
| Brain (50161) | 2.894 | 0.289 | 0.046 | **0.045** | 0.081 |
| Breast (45827) | 2.822 | 0.423 | **0.029** | **0.029** | 0.078 |
| Colorectal (44076) | 1.464 | 0.221 | **0.017** | 0.026 | 0.036 |
| Leukemia (28497) | 2.958 | 0.624 | 0.078 | 0.096 | **0.005** |
| Renal (53757) | 0.490 | 0.090 | 0.026 | 0.036 | **0.017** |
| MNIST | 65.107 | 37.896 | 9.935 | **9.885** | 2.35e5 |
| Fashion-MNIST | 35.235 | 1.955 | 0.613 | **0.612** | 2.80e5 |
| CIFAR10 | 26.598 | 1.276 | 0.858 | **0.850** | 1.63e5 |

Table 3: Evaluation Results on Average Geometric Distortion.

| Dataset | cMDS | Lower-MDS | Neuc-MDS | Neuc-MDS$^+$ |
|---|---|---|---|---|
| Random-Simplex | 1.049 | 1.049 | 1.010 | **1.004** |
| Euclidean Ball | 1.046 | 1.041 | **1.013** | 1.017 |
| Brain (50161) | 8.160 | 42.705 | **5.809** | 6.941 |
| Breast (45827) | 6.988 | 31.081 | **6.205** | 6.295 |
| Colorectal (44076) | 23.938 | 34.587 | **20.234** | 22.475 |
| Leukemia (28497) | **6.551** | 32.214 | 7.032 | 6.749 |
| Renal (53757) | 21.709 | 38.282 | **19.680** | 21.223 |
| MNIST | 1.119 | 1.104 | 1.064 | **1.063** |
| Fashion-MNIST | 1.135 | 1.096 | **1.068** | **1.068** |
| CIFAR10 | 1.129 | **1.109** | 1.121 | 1.118 |

**Performance on Dissimilarity Error.** In addition to STRESS as the primary metric, we also test the average distortion (i.e. multiplicative error) and scaled additive error on all datasets. For all metrics, a smaller value indicates a more favorable performance. Limited by space, we leave scaled additive error together with other observations in Appendix E. With $k$ as the target dimension, for synthetic datasets and images, we set $k = 100$, for genomics data, $k = 20$. The details are presented in Table 2.

The results[4] show that in terms of STRESS, Neuc-MDS and Neuc-MDS$^+$ outperform cMDS and Lower-MDS consistently by a large margin. SMACOF has comparable performance with Neuc-MDS in a couple data sets but can go out of bound in others. For average distortion, the genomics datasets differentiate different methods drastically while the rest datasets produce comparable results. Neuc-MDS$^+$ occasionally gives slightly higher STRESS than Neuc-MDS. Recall that both methods focus on optimizing for a lower bound of STRESS (in Theorem 2 and Theorem 6). This shows that $C_3$ may play a role in practice.

**Neuc-MDS Addresses 'Dimensionality Paradox'.** Dimensionality paradox of classical MDS refers to the observation that STRESS increases as the dimension goes up. When raising this concern, [38] proposes a Lower-MDS algorithm as mitigation. We show that Neuc-MDS and Neuc-MDS$^+$ address this issue even better. For Lower-MDS the target dimension cannot be larger than the number of positive eigenvalues. Our methods do not have this limitation. Figure 2 shows STRESS on Random-simplex and Renal. In Random-simplex, cMDS has an increasing STRESS with $k = 75 \sim 100$ then stops, and in Renal the STRESS keeps increasing. In contrast, Lower-MDS converges promptly while Neuc-MDS and Neuc-MDS$^+$ achieve even lower STRESS. Results on other datasets are in Appendix E.4.

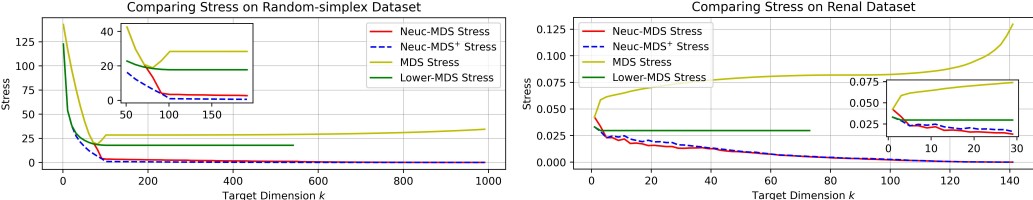

Figure 2: Neuc-MDS and Neuc-MDS$^+$ consistently produce lower STRESS on all dimensions. Lower-MDS has a shorter curve because the target dimension $k$ is limited to be smaller than the number of positive eigenvalues.

**Landmark MDS** Landmark MDS [12] is a heuristic to speed up classical MDS. One chooses a small number of landmarks and apply MDS on the landmarks first. The coordinates of the remaining points are obtained through a triangulation step with respect to the landmarks. We can use the same heuristic to speed up Neuc-MDS. On the random-simplex dataset, with only 25% points randomly chosen as landmarks the STRESS is only a factor of 1.0644 of the STRESS obtained by Neuc-MDS. If we use only 10% points as landmarks, the final STRESS is only 1.0898 of the STRESS of Neuc-MDS. This shows that Neuc-MDS can also be significantly accelerated using the landmark idea, achieving nearly the same STRESS.

# 7    Discussion and Conclusion

This paper presents an extension of classical MDS to non-Euclidean non-metric settings. We would like to mention a few future directions. Since we step out of the domain of Euclidean embedding, both the input dissimilarity matrix and the one obtained after dimension reduction can have negative values. Therefore if one would like to feed the output dissimilarity matrix to another data processing module that by default requires non-negative values, special care must be taken to address the negative values. In experiments, we discovered that Neuc-MDS$^+$ achieves similar stress as Neuc-MDS and produces much fewer negatives values in the output dissimilarity matrix (Appendix E.3). How to effectively use such dissimilarities in downstream learning and inference tasks would be a major future work. Note that the geometry of general bilinear forms is a largely unexplored territory.

## Acknowledgments and Disclosure of Funding

The authors acknowledge funding support through NSF IIS-2229876, DMS-2220271, DMS-2311064, CCF-2208663, CCF-2118953, and CRCNS-2207440.

---

[4]For the sake of clarity we scaled some results by a factor of $10^3$. We did so on all methods to ensure we did not engender bias on analysis. We provide the original results in Appendix E.3.

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

## A   Hollow Matrices and Relationship Between Gram and Squared Distance Matrices

We will prove the following proposition relating hollow symmetric matrices and squared "distance" matrices.

**Proposition 9.** *(1) Given any symmetric matrix $G = [g_{ij}]_{n \times n}$, there exists a symmetric bilinear form $\langle , \rangle$ on $\mathbb{R}^n$ and $n$ vectors $v_1, ..., v_n$ such that $g_{ij} = \langle v_i, v_j \rangle$.*

*(2) Given any hollow symmetric matrix $M = [m_{ij}]_{n \times n}$, there exists a symmetric bilinear form $\langle , \rangle$ on $\mathbb{R}^n$ and $n$ vectors $v_1, ..., v_n$ such that $m_{ij} = \langle v_i - v_j, v_i - v_j \rangle$ for all $i, j$.*

Note that the hollow condition $m_{ii} = 0$ is necessary.

*Proof.* Part (1) is a well known fact from linear algebra. We take $v_1, ..., v_n$ to be the standard basis of $\mathbb{R}^n$. From the symmetric matrix $G$, one defines the symmetric bilinear form by the formula $\langle u, v \rangle = u^T G v$ and vice versa.

To see part (2), consider the centered symmetric matrix $B = CMC$ where $C = I - \frac{J}{n}$ is the double centering matrix and $J$ is the matrix whose entries are 1. One can see that its $(i, j)$-th entry $b_{ij}$ of matrix $B$ is given by

$$b_{ij} = m_{ij} - \frac{1}{n} \sum_{k=1}^{n} (m_{ik} + m_{kj}) + \frac{1}{n^2} \sum_{k=1}^{n} \sum_{l=1}^{n} m_{kl}. \tag{9}$$

One can easily verify from the formula that the centering condition holds:

$$\sum_{k=1}^{n} b_{ik} = 0. \tag{10}$$

By part (1) of the proposition, one finds $n$ vectors $v_1, ..., v_n$ in $\mathbb{R}^n$ and a symmetric bilinear form $\langle , \rangle$ on $\mathbb{R}^n$ such that $-2b_{ij} = \langle v_i, v_j \rangle$. Now we claim that $m_{ij} = \langle v_i - v_j, v_i - v_j \rangle$, i.e., $-2m_{ij} = b_{ii} + b_{jj} - 2b_{ij}$.

Indeed, by dropping the constant term $\frac{1}{n^2} \sum_{k=1}^{n} \sum_{l=1}^{n} m_{kl}$ in (9), we have

$$b_{ii} + b_{jj} - 2b_{ij} \tag{11}$$

$$= m_{ii} - \frac{2}{n} \sum_{k=1}^{n} m_{ik} + m_{jj} - \frac{2}{n} \sum_{k=1}^{n} m_{kj} - 2m_{ij} + \frac{2}{n} \sum_{k=1}^{n} (m_{ik} + m_{kj}) \tag{12}$$

$$= -2m_{ij}. \tag{13}$$

The last step uses the fact that $M$ is a hollow matrix, i.e., $m_{ii} = m_{jj} = 0$. □

In summary, we call $G = [\langle v_i, v_j \rangle]$ the *Gram matrix* and $M = [\langle v_i - v_j, v_i - v_j \rangle]$ and *the squared distance matrix* of a symmetric bilinear form $\langle , \rangle$. Obviously, we can determine $M$ from $G$ by the formula $\langle v_i - v_j, v_i - v_j \rangle = \langle v_i, v_i \rangle + \langle v_j, v_j \rangle - 2 \langle v_i, v_j \rangle$, i.e., $m_{ij} = g_{ii} + g_{jj} - 2g_{ij}$. We can also recover the Gram matrix $G$ from $M$ from the double centering construction $CMC$ if the center of the vectors $v_1, ..., v_n$ is 0, i.e., $\sum_{i=1}^{n} v_i = 0$. To see this, given a squared distance matrix $M$ associated to $n$ vectors $v_1, .., v_n$, we can replace $v_i$'s by $v_i - w$ for a fixed vector $w$ without changing $M$. Now by taking $w = \frac{1}{n} \sum_{i=1}^{n} v_i$ to be the center of $v_1, ..., v_n$, we can normalize $\sum_{i=1}^{n} v_i = 0$. In this case, formula (9) states that

$$\langle v_i, v_j \rangle = -\frac{m_{ij}}{2} + \frac{1}{2n} \sum_{k=1}^{n} (m_{ik} + m_{kj}) - \frac{1}{2n^2} \sum_{k=1}^{n} \sum_{l=1}^{n} m_{kl}, \tag{14}$$

where $m_{ij} = \langle v_i - v_j, v_i - v_j \rangle$.

# B  Proofs of Error Analysis of Non-Euclidean MDS

In this section, we want to prove our main Theorem 6 for the error analysis of STRESS error. Note that Theorem 2 is a special case of Theorem 6 for $\Delta \boldsymbol{\lambda} \odot \boldsymbol{w} = 0$, which is exactly the case in MDS and Neuc-MDS with $\tilde{\lambda}$ given by $\boldsymbol{\lambda} \odot \boldsymbol{w}$.

**Theorem 6.** *Given a dissimilarity matrix $D \in \mathbb{R}^{n \times n}$ with eigenvalues $\boldsymbol{\lambda} \in \mathbb{R}^n$ of its Gram matrix $-CDC/2 = U \Lambda U^T$, for any rank-$k$ ($k \leq n$) diagonal matrix $\tilde{\Lambda} = \mathrm{Diag}(\tilde{\boldsymbol{\lambda}})$ with $\tilde{\boldsymbol{\lambda}} \in \mathbb{R}^n$ with $k$ nonzero entries only on coordinates given by some index $k$-set $W \subseteq [n]$, let $\boldsymbol{w} := \mathbf{1}_W \in \{0,1\}^n$ be the indicator vector of $W$, and let $\tilde{D}$ be the dissimilarity matrix reconstructed from $\tilde{X} = \tilde{\Lambda}^{1/2} U^T$. Then the STRESS error of $\tilde{D}$ can be expressed as: $\|\tilde{D} - D\|_F^2 = \tilde{C}_1 + \tilde{C}_2 + \tilde{C}_3$ with*

$$\tilde{C}_1 := 4 \left[ \bar{\boldsymbol{w}}^T \boldsymbol{\lambda}^{(2)} + \boldsymbol{w}^T \Delta \boldsymbol{\lambda}^{(2)} \right], \quad \tilde{C}_2 := 4 \left[ \bar{\boldsymbol{w}}^T \boldsymbol{\lambda} + \boldsymbol{w}^T \Delta \boldsymbol{\lambda} \right]^2, \quad \tilde{C}_3 := 2n \|(U \odot U)(\Delta \boldsymbol{\lambda})\|_F^2 - \frac{\tilde{C}_2}{2},$$

*where $\Delta \boldsymbol{\lambda} := \boldsymbol{\lambda} - \tilde{\boldsymbol{\lambda}}$. The first two terms, $\tilde{C}_1 + \tilde{C}_2$, as a lower bound of the STRESS, is minimized as*

$$4 \bar{\boldsymbol{w}}^T \boldsymbol{\lambda}^{(2)} + \frac{4 (\bar{\boldsymbol{w}}^T \boldsymbol{\lambda})^2}{1+k} \quad \text{with } \tilde{\boldsymbol{\lambda}} \text{ to be } \tilde{\boldsymbol{\lambda}}^* := \boldsymbol{\lambda} \odot \boldsymbol{w} + \frac{\bar{\boldsymbol{w}}^T \boldsymbol{\lambda}}{1+k} \boldsymbol{w}. \tag{7}$$

The proof will essentially follow the pipeline in [38]. Our results and proofs can be viewed as an extension to a more general family of problems. We first show that the STRESS error can be decomposed into three terms $\tilde{C}_1 + \tilde{C}_2 + \tilde{C}_3$. Then we analyze the approximate lower bound $\tilde{C}_1 + \tilde{C}_2$ and figure out the minimum.

We first introduce some notations and lemmas we need to use later. Recall that $C = I - \frac{1}{n} \mathbf{1}_n \mathbf{1}_n^T$ is the centering matrix. First note the following fact:

**Lemma 10.** *The $n$-dimensional vector $\mathbf{1}_n$ is an eigenvector with eigenvalue $0$ of the matrices $C, U$ and $\tilde{X}^T \tilde{X}$.*

*Proof.* By definition of $C = I - \frac{1}{n} \mathbf{1}_n \mathbf{1}_n^T$, it is easy to check $C \mathbf{1}_n = 0$. Also by definitions of $U$ and $\tilde{X}^T \tilde{X}$ we can get similar results. $\qquad \square$

As a corollary, we have the following lemma:

**Lemma 11.** $-C \tilde{D} C / 2 = C \tilde{X}^T \tilde{X} C = \tilde{X}^T \tilde{X}$.

*Proof.*
$$\tilde{D} = \mathrm{Diag}(G) \mathbf{1}_n - 2 \tilde{X}^T \tilde{X} + \mathbf{1}_n^T \mathrm{Diag}(G) \tag{15}$$
By Lemma 10, we have
$$-C \tilde{D} C / 2 = C \tilde{X}^T \tilde{X} C \tag{16}$$
By Lemma 10 again, we have
$$C \tilde{X}^T \tilde{X} C = \tilde{X}^T \tilde{X} \tag{17}$$
$$\qquad \square$$

Let $Q$ be a Householder reflector matrix defined as follows:

$$Q \triangleq I - \frac{2}{\boldsymbol{v}^T \boldsymbol{v}} \boldsymbol{v} \boldsymbol{v}^T \quad \text{with } \boldsymbol{v} = [1, \cdots, 1, 1 + \sqrt{n}]^T \tag{18}$$

For any symmetric matrix $A \in \mathbb{R}^{n \times n}$, let $\Phi(A) \in \mathbb{R}^{(n-1) \times (n-1)}, f(A) \in \mathbb{R}^{n-1}, \xi(A) \in \mathbb{R}$ be given by $QAQ$ through the following equation:

$$QAQ =: \begin{bmatrix} \Phi(A) & f(A) \\ f(A)^T & \xi(A) \end{bmatrix} \tag{19}$$

We need the following properties of $Q$:

**Lemma 12** ([32]). *For a symmetric matrix A, we have*

$$CAC = Q\begin{bmatrix}\Phi(A) & 0 \\ 0 & 0\end{bmatrix}Q. \tag{20}$$

**Lemma 13** ([21, 38]). *For any matrix $M$, we use $\mathrm{diag}(M) = (M_{i,i})$ to denote the column vector of $M$'s diagonal entries. For a symmetric hollow matrix $A$*

$$\begin{bmatrix} 2f(A) \\ \xi(A) \end{bmatrix} = \sqrt{n}Q\,\mathrm{diag}\left(Q\begin{bmatrix}\Phi(A) & 0 \\ 0 & 0\end{bmatrix}Q\right)$$

Now we are ready to prove our main theorem.

*Proof of Theorem 6.*

$$\|\tilde{D} - D\|_F^2 = \|Q(\tilde{D} - D)Q\|_F^2 \tag{21}$$
$$= \|\Phi(\tilde{D}) - \Phi(D)\|_F^2 + \|\xi(\tilde{D}) - \xi(D)\|_F^2 + 2\|f(\tilde{D}) - f(D)\|_F^2 \tag{22}$$

For the first term:

$$\|\Phi(\tilde{D}) - \Phi(D)\|_F^2 = \|Q\begin{bmatrix}\Phi(\tilde{D}) & 0 \\ 0 & 0\end{bmatrix}Q - Q\begin{bmatrix}\Phi(D) & 0 \\ 0 & 0\end{bmatrix}Q\|$$

$$\text{by Lemma 12} = \|C\tilde{D}C - CDC\|_F^2$$

$$= 4\|\frac{-C\tilde{D}C}{2} - \frac{-CDC}{2}\|_F^2$$

$$\text{by Lemma 11} = 4\|\tilde{X}^T\tilde{X} - \frac{-CDC}{2}\|_F^2 \tag{23}$$

$$= 4\|U(\Lambda - \tilde{\Lambda})U\|_F^2$$

$$= 4\|\Delta\boldsymbol{\lambda}\|_F^2$$

$$= 4\left[\bar{\boldsymbol{w}}^T\boldsymbol{\lambda}^{(2)} + \boldsymbol{w}^T\Delta\boldsymbol{\lambda}^{(2)}\right]$$

$$= \tilde{C}_1$$

For the second term:

From $0 = \mathrm{Tr}(D - \tilde{D}) = \mathrm{Tr}(\Phi(D) - \Phi(\tilde{D})) + (\xi(D) - \xi(\tilde{D}))$, we can get

$$\tilde{C}_2 := \|\xi(\tilde{D}) - \xi(D)\|_F^2 = (\xi(\tilde{D}) - \xi(D))^2 = 4(\sum_{i=1}^n \Delta\boldsymbol{\lambda}_i)^2 = 4[\bar{\boldsymbol{w}}^T\boldsymbol{\lambda} + \boldsymbol{w}^T\Delta\boldsymbol{\lambda}]^2 \tag{24}$$

For the third term:

By Lemma 13,

$$\begin{bmatrix} 2f(D) \\ \xi(D) \end{bmatrix} = \sqrt{n}Q\,\mathrm{diag}\left(Q\begin{bmatrix}\Phi(D) & 0 \\ 0 & 0\end{bmatrix}Q\right).$$

$$\begin{bmatrix} 2f(\tilde{D}) \\ \xi(\tilde{D}) \end{bmatrix} = \sqrt{n}Q\,\mathrm{diag}\left(Q\begin{bmatrix}\Phi(\tilde{D}) & 0 \\ 0 & 0\end{bmatrix}Q\right).$$

Taking the difference and then using a theorem on diagonalization and the Hadamard product from [24], we have:

$$\begin{bmatrix} 2f(D) - 2f(\tilde{D}) \\ \xi(D) - \xi(\tilde{D}) \end{bmatrix} = \sqrt{n}Q\,\mathrm{diag}\left(Q\begin{bmatrix}\Phi(D) - \Phi(\tilde{D}) & 0 \\ 0 & 0\end{bmatrix}Q\right)$$

$$\text{by Lemma 12} = \sqrt{n}Q\,\mathrm{diag}(C(D - \tilde{D})C)$$

$$= -2\sqrt{n}Q\,\mathrm{diag}(U\,\mathrm{Diag}(\Delta\boldsymbol{\lambda})U^T)$$

$$= -2\sqrt{n}Q(U \odot U)(\Delta\boldsymbol{\lambda})$$

Therefore,

$$\left\| \begin{bmatrix} 2f(D) - 2f(\tilde{D}) \\ \xi(D) - \xi(\tilde{D}) \end{bmatrix} \right\|_F^2 = 4n\|(U \odot U)(\Delta\boldsymbol{\lambda})\|_F^2 \tag{25}$$

Finally, we have that

$$2\|f(D) - f(\hat{D})\|_F^2 = 2n\|(U \odot U)(\Delta\boldsymbol{\lambda})\|_F^2 - \tilde{C}_2/2 = \tilde{C}_3 \tag{26}$$

The above arguments prove the first part of the Theorem.

Now we want to analyze the lower bound of

$$\tilde{C}_1 + \tilde{C}_2 = 4[\bar{\boldsymbol{w}}^T \boldsymbol{\lambda}^{(2)} + \boldsymbol{w}^T (\Delta\boldsymbol{\lambda})^{(2)}] + 4[\bar{\boldsymbol{w}}^T \boldsymbol{\lambda} + \boldsymbol{w}^T \Delta\boldsymbol{\lambda}]^2 \tag{27}$$

Denote $C := \bar{\boldsymbol{w}}^T \boldsymbol{\lambda}, Z := \boldsymbol{w}^T \Delta\boldsymbol{\lambda}$. By the Cauchy–Schwartz inequality, we know that

$$\boldsymbol{w}^T (\Delta\boldsymbol{\lambda})^{(2)} \geq \frac{1}{k}(\boldsymbol{w}^T \Delta\boldsymbol{\lambda})^2.$$

The equality is obtained when $\Delta\boldsymbol{\lambda}$ has the same value on all coordinates of $W$. Now check that

$$\boldsymbol{w}^T (\Delta\boldsymbol{\lambda})^{(2)} + [\bar{\boldsymbol{w}}^T \boldsymbol{\lambda} + \boldsymbol{w}^T \Delta\boldsymbol{\lambda}]^2 \geq \frac{1}{k}(\boldsymbol{w}^T \Delta\boldsymbol{\lambda})^2 + (C + Z)^2 \tag{28}$$

$$= \frac{1}{k}Z^2 + (C + Z)^2 \tag{29}$$

$$= \frac{1+k}{k}Z^2 + 2CZ + C^2 \tag{30}$$

$$\geq \frac{C^2}{1+k} \tag{31}$$

The last inequality achieves the lower bound with $Z = -\frac{k}{1+k}C$ through the standard analysis of univariate quadratic equations. Based on the above arguments, we have that

$$\tilde{C}_1 + \tilde{C}_2 \geq 4\bar{\boldsymbol{w}}^T \boldsymbol{\lambda}^{(2)} + \frac{4(\bar{\boldsymbol{w}}^T \boldsymbol{\lambda})^2}{1+k} \tag{32}$$

Since the lower bounds from the Cauchy–Schwartz inequality and the univariate quadratic equation are all tight, we can conclude that the final lower bound is tight and it can be achieved with

$$\Delta\boldsymbol{\lambda} = -\frac{\bar{\boldsymbol{w}}^T \boldsymbol{\lambda}}{1+k}\boldsymbol{w} \iff \tilde{\boldsymbol{\lambda}} = \boldsymbol{\lambda} \odot \boldsymbol{w} + \frac{\bar{\boldsymbol{w}}^T \boldsymbol{\lambda}}{1+k}\boldsymbol{w} \tag{33}$$

This finishes the proof. $\square$

## C  Proofs of Eigenvalue Selection Algorithm

*Proof of Theorem 3.* For any subset $S \subseteq \mathcal{L}$, define $F(S) = \sum_{\lambda \in \mathcal{L} \setminus S} \lambda^2 + \left(\sum_{\lambda \in \mathcal{L} \setminus S} \lambda\right)^2$. Note that this is the value we want to minimize (i.e., $C_1 + C_2$). Without loss of generality, we may assume that none of the chosen eigenvalues in $S$ is 0, since it does nothing to change $F(S)$.

We first prove the first claim of Theorem 3.

*Proof.* First, we show that an optimal solution $S$ of size $k$ exists. Assume we have some optimal solution $S$ of size less than $k$. Let $b \in \mathcal{L} \setminus S$. Then we have:

$$F(S \cup \{b\}) - F(S) = (-b^2) + (H(S) - b)^2 - H(S)^2 = -2bH(S) \tag{34}$$

Now it is clear that if $H(S)$ is 0, we can add any eigenvalue to $S$ and $F(S)$ would not change. If $H(S)$ is not 0, then adding an eigenvalue to $S$ that matches the sign of $H(S)$, would reduce $F(S)$ and the opposite would increase $F(S)$. We will continue to make use of this fact. Adding an eigenvalue that matches the sign of $H(S)$ is also always possible since

$H(S)$ is the sum of the eigenvalues we are choosing from. Thus, we can just continue to add eigenvalues to $S$ until it is of size $k$.

Next, assuming $S$ is of size $k$, we show that it must be the one as described in Theorem 3. We can prove this by contradiction. Without loss of generality, we may assume that $S$ contains a positive eigenvalue $b$ such that there exists some $a \in \mathcal{L} \setminus S$ satisfying $a > b$. First, we observe that since $S$ is optimal:

$$0 \leq F(S \setminus \{b\}) - F(S) = b^2 + (H(S) + b)^2 - H(S)^2 = 2bH(S) + 2b^2 \tag{35}$$

which reduces to $H(S) \geq -b$. Then, we have:

$$F(S \cup \{a\} \setminus \{b\}) - F(S) = (b^2 - a^2) + (H(S) + b - a)^2 - H(S)^2 = (b - a)(2b + 2H(S)) \tag{36}$$

We know that $b - a < 0$ and $b > 0$, $H(S) \geq -b$, so we get that the expression is negative. Thus, $S \cup \{a\} \setminus \{b\}$ has a lower value under $F$ and contradicts our assumption that $S$ is optimal. $\qquad\square$

Let $S$ be the optimal solution from Theorem 3. Let $T$ be the solution obtained by EV-Selection. We will show that $F(S) = F(T)$. We use contradiction again. Without loss of generality, we assume that $S$ has more positive eigenvalues than $T$. Now consider $H(S \cap T)$. From a similar argument used in the proof of Theorem 3 (1), due to $S \setminus T$ containing only positive eigenvalues, $H(S \cap T) \geq 0$. Now if $H(S \cap T) = 0$, we know that $S \setminus T$ contains one eigenvalue, — otherwise, removing a positive eigenvalue from $S$ would decrease $F(S)$. Then, we know that $F(S) = F(T)$ and that is a contradiction. If $H(S \cap T) > 0$, then we have reached a contradiction in the definition of $T$ because EV-Selection will only choose eigenvalues in $T \setminus S$ when all negative eigenvalues in $T \cap S$, $Q$, has been chosen. However, for all subsets $P$ of positive eigenvalues of $T$, $H(P \cup Q) > 0$, so no eigenvalue in $T \setminus S$ will ever be chosen by EV-Selection. $\qquad\square$

---

**Algorithm 3**: Eigenvalue Selection for Generalized Neuc-MDS

---

**Input**: Sorted eigenvalues $\boldsymbol{\lambda}$, integer $k$.
**Output**: Set $S$ of selected eigenvalues
Let $\mathcal{L}$ be the set of all eigenvalues $\boldsymbol{\lambda}$, $S = \emptyset$
**while** $|S| < k$ **do**
    $T_+ = S \cup \arg\max_{\lambda \in \mathcal{L} \setminus S, \lambda > 0}$
    $T_- = S \cup \arg\max_{\lambda \in \mathcal{L} \setminus S, \lambda < 0}$
    $A_1 = \sum_{\lambda \in \mathcal{L} \setminus T_+} \lambda^2 + \frac{1}{|T_+| + 1} \left( \sum_{\lambda \in \mathcal{L} \setminus T_+} \lambda \right)^2$
    $A_2 = \sum_{\lambda \in \mathcal{L} \setminus T_-} \lambda^2 + \frac{1}{|T_-| + 1} \left( \sum_{\lambda \in \mathcal{L} \setminus T_-} \lambda \right)^2$
    **if** $A_1 < A_2$ **then**
        $S = T_+$
    **else**
        $S = T_-$
    **end**
**end**

---

*Proof of Proposition 8.* For any subset $S \subseteq \mathcal{L}$, define $F(S) = \sum_{\lambda \in \mathcal{L} \setminus S} \lambda^2 + \frac{1}{|S| + 1} \left( \sum_{\lambda \in \mathcal{L} \setminus S} \lambda \right)^2$. Note that this is the value we want to minimize (i.e., $C_1 + \frac{1}{|S| + 1} C_2$). Without loss of generality, we may assume that none of the chosen eigenvalues in $S$ is 0, since it does nothing to change $F(S)$.

We first prove the first claim in Proposition 8.

*Proof.* First, we show that an optimal solution $S$ of size $k$ exists. Assume we have some optimal solution $S$ of size less than $k$. Let $b \in \mathcal{L} \setminus S$. Then we have:

$$F(S \cup \{b\}) - F(S) = (-b^2) + \frac{1}{|S| + 2}(H(S) - b)^2 - \frac{1}{|S| + 1}H(S)^2 \leq -\frac{2}{|S| + 1}bH(S) \tag{37}$$

Now it is clear that if $H(S)$ is 0, adding any eigenvalue to $S$ would not increase $F(S)$. If $H(S)$ is not 0, then adding an eigenvalue to $S$ that matches the sign of $H(S)$, would reduce $F(S)$. We will continue to make use of this fact. Adding an eigenvalue that matches the sign of $H(S)$ is also always possible since $H(S)$ is the sum of the eigenvalues we are choosing from. Thus, we can just continue to add eigenvalues to $S$ until it is of size $k$.

Next, we will show there must be an $S$ as described in Proposition 8. We can prove this by contradiction. Without loss of generality, we may assume some optimal $S$ contains a positive eigenvalue $b$ such that there exists some $a \in \mathcal{L} \setminus S$ satisfying $a > b$. We can directly observe since $S$ is optimal

$$0 \leq F(S \cup \{a\} \setminus \{b\}) - F(S) = (b^2 - a^2) + \frac{1}{|S| + 1}((H(S) + b - a)^2 - H(S)^2)$$

$$= \frac{1}{|S| + 1}(b - a)((|S| + 2)b + |S|a + 2H(S))$$

Since we know $b - a < 0$, we get that $(|S| + 2)b + |S|a + 2H(S) \leq 0$. First, for the case where it is equal to 0, we can just take out $b$ and put in $a$. If $S$ still does not have the greatest positive eigenvalues, we can repeat our analysis and examine another candidate positive eigenvalue to be replaced. In the case that it is strictly negative, then we have $H(S) < 0$ since $|S|, b, a > 0$. Then there must exist some negative eigenvalue $c$ that has not been chosen and if we replace $a$ with $c$ in our analysis, we get that $b - c > 0$ and $(|S| + 2)b + |S|c + 2H(S) < 0$ which gives $F(S \cup \{c\} \setminus \{b\}) < F(S)$ which contradicts the optimality of $S$. □

Let $S$ be the optimal solution from Proposition 8. Let $T$ be the solution obtained by ?? 3. We will show that $F(S) = F(T)$. Note that now that we know $|S| = k$, we now define $F = \sum_{\lambda \in \mathcal{L} \setminus S} \lambda^2 + \frac{1}{k+1}\left(\sum_{\lambda \in \mathcal{L} \setminus S} \lambda\right)^2$. We use contradiction again. Without loss of generality, we assume that $S$ has more positive eigenvalues than $T$. Let $a$ be the largest eigenvalue in magnitude from $T \setminus S$. Let $T'$ be the set chosen by ?? 3 right before choosing $a$. Then let $b$ be the largest eigenvalue in $S \setminus T'$. By definition of $T$, we know that:

$$0 \geq F(T' \cup \{a\}) - F(T' \cup \{b\}) = (b^2 - a^2) + \frac{1}{k+1}((H(T') - a)^2 - (H(T') - b)^2)$$

$$= (b^2 - a^2) + \frac{1}{k+1}(a - b)(a + b - 2H(T'))$$

Now let's compare that with:

$$F(S \cup \{a\} \setminus \{b\}) - F(S) = (b^2 - a^2) + \frac{1}{k+1}((H(S) + b - a)^2 - (H(S) + b - b)^2)$$

$$= (b^2 - a^2) + \frac{1}{k+1}(a - b)(a + b - 2(H(S) + b))$$

Now clearly $H(S) + b \leq H(T')$. In the case that $H(S) + b = H(T')$, we clearly have $S \cup \{a\} \setminus \{b\} = T$, so we get $F(S) = F(T)$. In the case that $H(S) + b < H(T')$, we also know that $a - b < 0$, so we get $F(S \cup \{a\} \setminus \{b\}) - F(S) < F(T' \cup \{a\}) - F(T' \cup \{b\}) \leq 0$. That violates the fact that $S$ is an optimal set, so we have a contradiction. □

## D   Computations on Random Matrices

We compare the classical multidimensional scaling and our non-Euclidean multidimensional scaling under the random matrix context, and give the proof of Theorem 5.

We start with a symmetric random matrix $B \in \mathbb{R}^{n \times n}$ as in Proposition 4, and want to select $k$ eigenvalues by using the classical and non-Euclidean multidimensional scaling.

We start with classical multidimensional scaling. To select the largest $k$ eigenvalues, we need to select all eigenvalues greater than $(2 - r)\sqrt{n}\sigma$ such that

$$\frac{k}{n} = \frac{1}{2\pi\sigma^2} \int_{(2-r)\sigma}^{2\sigma} \sqrt{4\sigma^2 - x^2}\, dx = \frac{1}{2} - \frac{1}{\pi}\arcsin\left(1 - \frac{r}{2}\right) - \frac{1}{\pi}\left(1 - \frac{r}{2}\right)\sqrt{r \cdot \left(1 - \frac{r}{4}\right)}. \quad (38)$$

There is not an explicit way to write $r$ as a function of $k$. For a fixed $r$, since we want to drop all eigenvalues smaller than $(2-r)\sqrt{n}\sigma$, we have

$$
\begin{aligned}
&e_C \\
&= \Sigma_{\lambda \in \mathcal{L}\setminus S}\lambda^2 + (\Sigma_{\lambda \in \mathcal{L}\setminus S}\lambda)^2 \\
&= n^2 \cdot (\Sigma_{\lambda \in \mathcal{L}\setminus S}(\frac{\lambda}{\sqrt{n}})^2 \cdot \frac{1}{n}) + n^3 \cdot (\Sigma_{\lambda \in \mathcal{L}\setminus S}\frac{\lambda}{\sqrt{n}} \cdot \frac{1}{n})^2 \\
&= n^2 \cdot \frac{1}{2\pi\sigma^2}\int_{-2\sigma}^{(2-r)\sigma} x^2\sqrt{4\sigma^2 - x^2}dx + n^3 \cdot (\frac{1}{2\pi\sigma^2}\int_{-2\sigma}^{(2-r)\sigma} x\sqrt{4\sigma^2 - x^2}dx)^2 \\
&= n^2\sigma^2\Big[\frac{1}{2} + \frac{1}{\pi}\arcsin(1 - \frac{r}{2}) + \frac{1}{\pi}(1 - \frac{r}{2})(1 - 2r + \frac{r^2}{2})\sqrt{r(1 - \frac{r}{4})} + \frac{16n}{9\pi^2}r^3(1 - \frac{r}{4})^3\Big].
\end{aligned}
$$ (39)

For non-Euclidean multidimensional scaling, to select the $k$ eigenvalues with largest magnitude, we need to select all eigenvalues with magnitude greater than $(2 - r)\sqrt{n}\sigma$ such that

$$
\begin{aligned}
\frac{k}{n} &= \frac{1}{2\pi\sigma^2}(\int_{(2-r)\sigma}^{2\sigma}\sqrt{4\sigma^2 - x^2}dx + \int_{-2\sigma}^{-(2-r)\sigma}\sqrt{4\sigma^2 - x^2}dx) \\
&= 1 - \frac{2}{\pi}\arcsin(1 - \frac{r}{2}) - \frac{2}{\pi}(1 - \frac{r}{2})\sqrt{r \cdot (1 - \frac{r}{4})}.
\end{aligned}
$$ (40)

For a fixed $r$, since we want to drop all eigenvalues with magnitude smaller than $(2-r)\sqrt{n}\sigma$, we have

$$
\begin{aligned}
e_N &= \Sigma_{\lambda \in \mathcal{L}\setminus S}\lambda^2 + (\Sigma_{\lambda \in \mathcal{L}\setminus S}\lambda)^2 \\
&= n^2 \cdot (\Sigma_{\lambda \in \mathcal{L}\setminus S}(\frac{\lambda}{\sqrt{n}})^2 \cdot \frac{1}{n}) \\
&= n^2 \cdot \frac{1}{2\pi\sigma^2}\int_{-(2-r)\sigma}^{(2-r)\sigma} x^2\sqrt{4\sigma^2 - x^2}dx \\
&= n^2\sigma^2\Big[\frac{2}{\pi}\arcsin(1 - \frac{r}{2}) + \frac{2}{\pi}(1 - \frac{r}{2})(1 - 2r + \frac{r^2}{2})\sqrt{r(1 - \frac{r}{4})}\Big].
\end{aligned}
$$ (41)

We first consider the case that $k = o(n)$, i.e. we want to reduce the dimension from $n$ to a much smaller $k$. Under this assumption, $r$ is a very small positive number. In the classical case, by applying the Taylor series to equation (38), we get $r \approx (\frac{3\pi k}{2n})^{\frac{2}{3}}$. Then we plug it into equation (39) to get

$$
e_C \approx n^2\sigma^2(1 + \frac{4k^2}{n} - \frac{4k}{n}).
$$

Similarly, in the non-Euclidean case, we apply the Taylor series to equation (40), we get $r \approx (\frac{3\pi k}{4n})^{\frac{2}{3}}$. Then we plug it into equation (41) to get

$$
e_N \approx n^2\sigma^2(1 - \frac{4k}{n}).
$$

This finishes the proof of Theorem 5 (1).

Now we turn to the case that $k = cn$ for a constant $c \in [0, 1]$. We first solve $c = \frac{k}{n}$ in equations (38) and (40) to get the corresponding $r$, then plug the $r$-values in equations (39) and (41) to get the corresponding $e_C$ and $e_N$, respectively. Since equations (38) and (40) can not be solved explicitly, we can only get numerical values of $e_C$ and $e_N$. For $c \in (0, 0.5]$, numerical values of $e_C$ and $e_N$ as shown in Table 4. For $c \in [0.5, 1]$, the classical multidimensional scaling stabilizes, with $e_C \approx (0.5 + 0.1801 \cdot n)n^2\sigma^2$. Meanwhile, the error $e_N$ for non-Euclidean multidimensional scaling decreases to zero as $c$ increases to 1. More precisely, if $c = 1 - \epsilon$ with very small $\epsilon > 0$, equations (40) and (41) give

$$
e_N \approx \frac{\pi^2}{12}\epsilon^3 n^2\sigma^2.
$$

Some numerical values of $e_N$ with $c \in [0.5, 1]$ are shown in Table 5. This finishes the proof of Theorem 5 (2).

| $c$ | 0.05 | 0.1 | 0.15 |
|---|---|---|---|
| $e_C/(n^2\sigma^2)$ | $0.8432 + 0.0078 \cdot n$ | $0.7322 + 0.0265 \cdot n$ | $0.6513 + 0.0512 \cdot n$ |
| $e_N/(n^2\sigma^2)$ | 0.8278 | 0.6864 | 0.5666 |
| $c$ | 0.2 | 0.25 | 0.3 |
| $e_C/(n^2\sigma^2)$ | $0.5933 + 0.0785 \cdot n$ | $0.5531 + 0.1055 \cdot n$ | $0.5269 + 0.1304 \cdot n$ |
| $e_N/(n^2\sigma^2)$ | 0.4644 | 0.3771 | 0.3027 |
| $c$ | 0.35 | 0.4 | 0.45 |
| $e_C/(n^2\sigma^2)$ | $0.5112 + 0.1512 \cdot n$ | $0.5033 + 0.1670 \cdot n$ | $0.5004 + 0.1768 \cdot n$ |
| $e_N/(n^2\sigma^2)$ | 0.2397 | 0.1866 | 0.1425 |

Table 4: This table shows the error terms $e_C$ and $e_N$ of different choices of $c \in (0, 0.5)$ with $c = k/n$, normalized by dividing $n^2\sigma^2$.

| $c$ | 0.5 | 0.55 | 0.6 | 0.65 | 0.7 |
|---|---|---|---|---|---|
| $e_N/(n^2\sigma^2)$ | 0.1063 | 0.0770 | 0.0537 | 0.0358 | 0.0225 |
| $c$ | 0.75 | 0.8 | 0.85 | 0.9 | 0.95 |
| $e_N/(n^2\sigma^2)$ | 0.0130 | 0.0066 | 0.0028 | 0.0008 | 0.0001 |

Table 5: This table shows the error term $e_N$ of different choices of $c \in [0.5, 1)$ with $c = k/n$, normalized by dividing $n^2\sigma^2$.

# E    More Details and Results on Experiments

In this appendix we fill the missing details in the experiment section. Our experiments are implemented with Intel Core i9 CPU of 32GB memory, no GPU is required and the execution time is no longer than 30 seconds. In the rest of this section, we first introduce the generation process of synthetic datasets in Appendix E.1; Next we define the evaluation metrics and perturbation approaches in Appendix E.2; More results on dissimilarities regarding the scaled additive error, negative distances and negative eigenvalues selected, are demonstrated in Appendix E.3. With respect to the *dimensionality paradox*, we provide illustrations similar to Figure 2 on other datasets in Appendix E.4.

## E.1    Synthetic Datasets Generation

The *random-simplex* dataset is generated with the idea of creating a simplex with one added dimension that generates all of the variance in the the distances such that the added dimension corresponds with a negative eigenvalue. This way, we should have a dimension with a large negative eigenvalue that is impactful on the stress of the embedding.

Specifically, this dataset was generated with points $x_1, ... x_{1000}$, each point having 1000 dimensions. For each point $x_i$, the first 100 coordinates were chosen uniformly randomly between 0 and .01. The next 899 coordinates were chosen uniformly randomly between 0 and $\sqrt{0.5/899}$. The last coordinate is $i \times 0.3/1000$. Then, the distance between $x_i$ and $x_j$ was found in the following way:

$$d(x_i, x_j) = \sqrt{\sum_{k=1}^{100}(x_i(k) - x_j(k))^2 - \sum_{k=101}^{1000}(x_i(k) - x_j(k))^2}$$

where $x_i(k)$ is the $k$th coordinate of $x_i$. The first 999 coordinates essentially form a simplex, because selecting a large number of coordinates randomly from the same distribution causes all of the distances to converge to a normal distribution with low variance. Thus, all of the distances between points using just the first 999 coordinates are very similar. Then, the last coordinate has "large" discrepancies between all of the points, causing most of the differences in the distances.

The *Euclidean-ball* metric was inspired by [49]. The set of distances between objects in space does not follow triangle inequality, so it becomes a non-euclidean dissimilarity measure. The particular dataset we use generates the distances by first randomly selecting 1000 points

uniformly in a 10 dimensional hypercube of length 100. Each of these points is now the center of a ball. Then, for each ball, with probability 0.9, we select a radius uniformly randomly between 0 and 5. With probability 0.1, we let the radius be 0.8 times the distance between the center of this ball to the closest other ball. This closest other ball is based on the radius of the other ball if it has already been decided. The final distance matrix is just filled by the distances between the 1000 balls. In this way, we attempt to create some balls with large radius, so that the triangle inequality is heavily violated, therefore causing the dataset to deviate more from the Euclidean setting.

## E.2  Evaluation and Perturbation Metrics

In addition to STRESS, our evaluation metric include scaled additive error and average geometric distortion. Scaled additive error by definition is allowing scaling of the distance matrix of our embedding before calculating stress. To do this, we first ran the embedding method (e.g., cMDS, Neuc-MDS, or other methods) on the given dissimilarity squared matrix $D$ to get the output dissimilarity squared matrix $\tilde{D}$. Then, we would flatten both $D$ and $\tilde{D}$ to vectors, and project $D$ onto the line through $\tilde{D}$ to get $E$. Then we calculate $\|D - E\|$ to get scaled additive error. Clearly, this is equivalent to allowing scaling of $\tilde{D}$ to find the minimum stress. For average geometric distortion, we again input $D$ to get $\tilde{D}$, but this time, we evaluate the distortion of each dissimilarity by dividing entries of $\sqrt{D}$ by entries of $\sqrt{\tilde{D}}$. Since distortion is not well defined when dissimilarities become complex, we skip those. Then, with the remaining valid distortions, we scale the distortions so that there's an equal number of them greater than 1 and less than 1, then for those that remain less than 1, we take their reciprocal. Then, we finally take the geometric average of all of these distortions. In this way, we basically again allowed scaling of the dissimilarities to find the minimum possible average distortion.

Next, we had to decide on metrics to use for the image datasets. We took the following 3 metrics from [38]. For the first metric, the distance between two images is first finding the Euclidean distance between their coordinate representations, and then adding Gaussian noise to the distances. Because the noise is added directly onto the distances, the end result is highly likely to be non-Euclidean. For the second metric, we build a $k$ nearest neighbor graph of the images based on their Euclidean distances. Since the end result is a graph structure, the shortest path distances are also highly likely to be non-Euclidean. For the third metric, we randomly removed entries from the coordinate representation of the images. Then, the distance between two images is decided by taking the Euclidean distance between coordinates that both images still included. Here, the Euclidean property breaks down again because triangle inequality would no longer have to hold. In Table 2, we choose the $k$-NN metric with $k = 2$. In the next Appendix section, we show results for higher $k$ and other metrics applied to the image datasets. As for genomics datasets, we use the entropic affinities commonly used in t-SNE methods. The main idea is to apply an adaptive kernel, whose bandwidth depends on a parameter termed as perplexity, to pairwise data entries.

## E.3  More Results on Dissimilarity Error

In addition to STRESS and average distortion already reported in Table 2, we further show the third metric: scaled additive error as defined in Appendix E.2. Note that our algorithms may produce negative distances, we also report number of negative distances and number of negative eigenvalues selected. All original results (without scaling) on 10 datasets are demonstrated in Table 6. Recall we mentioned the results in Table 2 contains some scaling on the synthetic data and images, we also provide the original values in Table 7.

On scaled additive error, our proposed methods still consistently outperform cMDS and Lower-MDS considerably. Thus, we have shown that on all metrics our methods yield more favorable results. It is interesting to observe that Neuc-MDS$^+$ produces much fewer negative distances than Neuc-MDS, though the number of negative eigenvalues selected are quite close. Further, two methods have similar performance on STRESS and other metrics. Therefore in practice, if there is a considerable concern on negative distances, Neuc-MDS$^+$ should become a better choice.

Table 6: Original Evaluation Results on All Datasets for Lower-MDS (L-MDS), Neuc-MDS (N-MDS) and Neuc-MDS$^+$ (N-MDS$^+$). Metrics include scaled additive error, number of negative distances and number of negative eigenvalues selected.

| Dataset | Scaled Additive Error | | | | # Neg Distances | | # Neg $\lambda$ Selected | |
|---|---|---|---|---|---|---|---|---|
| | cMDS | L-MDS | N-MDS | N-MDS$^+$ | N-MDS | N-MDS$^+$ | N-MDS | N-MDS$^+$ |
| Random-simplex | 17758 | 17760 | 3415 | **1392** | 0 | 0 | 8 | 1 |
| Euclidean-ball | 14909 | 13093 | **3132** | 3631 | 798 | 523 | 90 | 87 |
| Brain (50161) | 0.131 | 0.132 | 0.063 | **0.062** | 5539 | 1081 | 8 | 9 |
| Breast (45827) | 0.039 | 0.039 | 0.016 | **0.015** | 9024 | 136 | 8 | 9 |
| Colorectal (44076) | 0.027 | 0.027 | **0.012** | 0.014 | 12041 | 1940 | 6 | 8 |
| Leukemia (28497) | 0.031 | 0.031 | **0.021** | 0.023 | 32705 | 2102 | 8 | 10 |
| Renal (53757) | 0.018 | 0.018 | **0.013** | 0.014 | 6650 | 713 | 7 | 9 |
| MNIST | 7592 | 6126 | 3148 | **3.135** | 1006 | 68 | 42 | 43 |
| Fashion-MNIST | 6153 | 4421 | 2.472 | **2.470** | 526 | 8 | 41 | 41 |
| CIFAR10 | 3880 | 3564 | 2870 | **2.842** | 2968 | 201 | 43 | 44 |

Table 7: Original Evaluation Results on All Datasets. Metrics include STRESS and average geometric distortion

| Dataset | STRESS | | | | Average Geometric Distortion | | | |
|---|---|---|---|---|---|---|---|---|
| | cMDS | L-MDS | N-MDS | N-MDS$^+$ | cMDS | L-MDS | N-MDS | N-MDS$^+$ |
| Random-simplex | 28.376 | 17.760 | 3.433 | **1.392** | 1.049 | 1.049 | 1.010 | **1.004** |
| Euclidean-ball | 19.229 | 13.154 | **3.346** | 3.676 | 1.046 | 1.041 | **1.013** | 1.017 |
| Brain (50161) | 0.538 | 0.170 | 0.068 | **0.067** | 8.160 | 42.705 | **5.809** | 6.941 |
| Breast (45827) | 0.168 | 0.065 | **0.017** | **0.017** | 6.988 | 31.081 | **6.205** | 6.295 |
| Colorectal (44076) | 0.121 | 0.047 | **0.013** | 0.016 | 23.938 | 34.587 | **20.234** | 22.475 |
| Leukemia (28497) | 0.172 | 0.079 | **0.028** | 0.031 | 6.551 | 32.214 | 7.032 | 6.749 |
| Renal (53757) | 0.070 | 0.030 | **0.016** | 0.019 | 21.709 | 38.282 | **19.680** | 21.223 |
| MNIST | 25.516 | 6.156 | 3.152 | **3.144** | 1.119 | 1.104 | 1.064 | **1.063** |
| Fashion-MNIST | 18.771 | 4.422 | 2.475 | **2.473** | 1.135 | 1.096 | **1.068** | **1.068** |
| CIFAR10 | 16.309 | 3.572 | 2.930 | **2.916** | 1.129 | **1.109** | 1.121 | 1.118 |

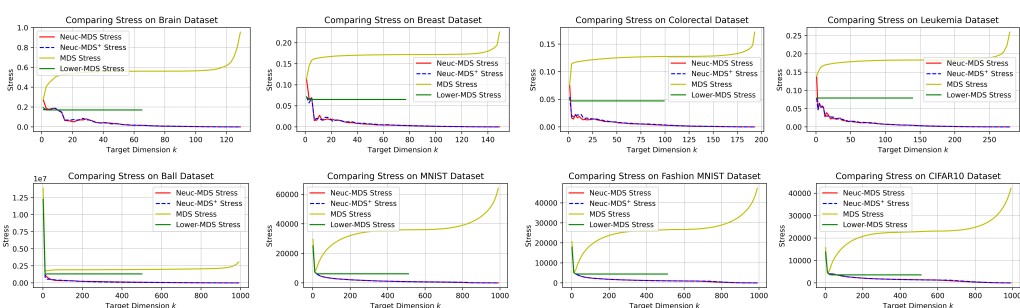

Figure 3: Dimensionality Paradox on Eight Datasets.

Until now all results reported on images are using $k$NN metric with $k = 2$ to produce non-Euclidean dissimilarities. There could be other perturbations for the same purpose, as mentioned in Appendix E.2. We further test with $k = 10$ and the other two metrics: adding noise and random removal on the STRESS. We add noise sampled from Gaussian distribution with variance as the maximum instance-wise difference scaled by 500; For random removal, we skip 50 images for each. The results are shown in Table 8.

Table 8: Evaluation Results on Image Datasets with other perturbation metrics.

| Dataset | STRESS | | | | # Neg distances | | # Neg $\lambda$ Selected | |
| --- | --- | --- | --- | --- | --- | --- | --- | --- |
| | cMDS | L-MDS | N-MDS | N-MDS$^+$ | N-MDS | N-MDS$^+$ | N-MDS | N-MDS$^+$ |
| MNIST ($k = 10$) | 10080 | 1979 | **1827** | 1830 | 554 | 63 | 42 | 41 |
| Fashion ($k = 10$) | 10249 | 2112 | 1936 | **1915** | 403 | 84 | 38 | 40 |
| CIFAR10 ($k = 10$) | 9032 | **1881** | 2309 | 2329 | 3236 | 996 | 39 | 37 |
| MNIST (noise) | 1.497e9 | 1.499e9 | 1.497e9 | 1.140e9 | 0 | 0 | 0 | 7 |
| Fashion (noise) | 4.815e9 | 4.130e9 | 3.382e9 | 3.327e9 | 0 | 0 | 2 | 14 |
| CIFAR10 (noise) | 2.303e10 | 2.116e9 | 2.303e9 | 1.848e9 | 0 | 0 | 0 | 9 |
| MNIST (missing) | 3.281e8 | 1.010e8 | 3.281e8 | 1.010e8 | 0 | 0 | 0 | 0 |
| Fashion (missing) | 5.165e8 | 1.726e8 | 5.165e8 | 1.726e8 | 0 | 0 | 0 | 0 |
| CIFAR10 (missing) | 2.132e9 | 5.756e8 | 2.132e9 | 5.756e8 | 0 | 0 | 0 | 0 |

## E.4 More Results on Dimensionality Paradox

The purpose of this section is to show the consistent performance of Neuc-MDS and Neuc-MDS$^+$ on all datasets, with respect to mitigating the dimensionality paradox issue. Therefore, in addition to Figure 2, we provide the same plots for other datasets, and the parameters all follow the main setup. Figure 3 gives a clear illustration that Neuc-MDS and Neuc-MDS$^+$ always have lower STRESS than cMDS and Lower-MDS as dimension grows.

