# OpenReview forum: "Neuc-MDS: Non-Euclidean Multidimensional Scaling Through Bilinear Forms"
_NeurIPS.cc/2024/Conference — NeurIPS 2024 poster_

### Official Review · Reviewer_EWdH · 2024-07-11

**Soundness:** 3
**Presentation:** 3
**Contribution:** 3
**Rating:** 6
**Confidence:** 2

**Summary:**

In this paper, the authors propose a variant of MDS that is able to deal with data with a non-Euclidean structure. The main idea is to generalize the inner product to a bilinear form, hence admitting some relationships corresponding to "negative eigenvalues," interpreted as in an inner product form $u^TAv$. Although a closed-form solution is not possible, the authors optimize over a lower bound. The paper ends with empirical evaluations both on synthetic and real data, with promising results.

**Strengths:**

The paper is well written in general. The introduction and motivation are very clear. Although most of the bounds and some ideas are already in [38], the idea of including negative eigenvalues and its implementation is interesting.

Another strength is the availability of the code.

**Weaknesses:**

The introduction and problem definition are quite detailed (which is good), but I think that some of these facts may be considered well-known, like basic linear algebra facts. This could be used to include more specific material with the page restriction.

I also think that the experimental section needs a subsection to interpret the embeddings. Throughout the paper (and supplementary material) there are numerical results reported, but the embeddings are never plotted. At least for some toy example.

**Questions:**

I have a couple of questions related to real/complex values.

In line 99. why real/imaginary-valued dissimilarity when in the next line it reads $D\in \mathbb{R}^{n \times n}$ ?

In equation (3). Then $X$ is complex (since $\sqrt{\Lambda}$ is). Because in the problem formulation you are looking for real embeddings.

Also related to that. If $X$ is complex, in the equation at the end of that page (page 4), shouldn't it be the complex conjugate instead of only the transpose?

Somehow related to these last two points. Does it make sense to include the sign of the eigenvalue in Diag(w) (and therefore $w$ can have values in $\{0,1,-1\}$, and the (squared root) absolute value of the eigenvalues in $\sqrt{\Lambda}$?
Something like the Generalized RDPG in "A statistical interpretation of spectral embedding: the generalised random dot product graph" by Rubin-Delanchy et al.

In line 303:  "all dataset are non-metric", but then in Table 1 there's a column indicating wether the dataset is metric.

Minor comments:
- In equation (1), use \left( \right)
- Line 259:  "The we ask"

**Limitations:**

Although there is no specific section for this, some limitations are commented on throughout the manuscript.

---

> ### Author Rebuttal · Authors · 2024-08-07
>
> We sincerely thank you for your careful reading and valuable feedback. In the following we address the concerns.
>
> - ### Regarding complex values
> In Line 99, we apologize for the confusion. $D$ is the squared dissimilarity matrix in general. We will emphasize it in the revision.
>
> Regarding the complex (pure imaginary) entries in $X$, first we want to clarify that, these imaginary entries do not lie in traditional complex plane as one would expect. The reason is that the inner product (bilinear form) we use here is not the same as the traditional complex inner product (taking conjugate) defined for a complex plane.  Therefore, the geometric meaning is totally different. This also partially answers the question why we could not use the complex conjugate in the equation at the end of page 4. The complex conjugate will produce a wrong dissimilarity matrix which cannot reconstruct the original one.
>
> In the old version paper, we allow complex values in $X$ in order to simplify the mathematic expressions (the sign of eigenvalues will be absorbed). However, as observed now, we found it brings more confusion. In the revised vision later, we would only use real valued X together with a signed diagonal matrix $A=sign(\lambda)$. More precisely,  let $X=\sqrt{diag{|\lambda|\odot w}} \cdot U^T$, and then use bilinear form $f(u, v)=u^TAv$ with the matrix $A$ as a diagonal matrix with $k$ non-zero values in $\{+1, -1\}$, with the sign matching the sign of the eigenvalues selected in $w$. Then $\hat{D}_{ij}=f(X_i-X_j, X_i-X_j)=(X_i-X_j)^TA(X_i-X_j)$.
>
> Regarding the sign and absolute eigenvalues, yes, it contains important information. For our distance reconstruction tasks, we weighted eigenvectors by the square root of eigenvalues, together with signed diagonal matrix of eigenvalues. In other applications, there are different ways to combine these information.
>
> The referenced paper of the work regarding Generalized RDPG is very interesting and highly related to our work. In general, our settings work for any symmetric dissimilarity (hollow) matrices, and one important motivation is to study the graph or network structure which is usually far from Euclidean geometry. One example mentioned in that paper about stochastic block models with cross-block probability higher than in-block probability,  $B_{1,2} > B_{1,1}, B_{2,2}$, might be highly related to the hyberbolic geometry which is one of our motivating examples to study the general bilinear form with both positive and negative eigenvalues. Also, the way of visualization and interpretations of negative eigenvalues also inspire us a lot. We believe the study in that paper could be a good example and support of our work. We will include some discussions in the revised version.
>
> In line 303, there is a typo. It should be “all dataset are non-Euclidean or non-metric”. Non-metric means the dissimilarity matrix even does not satisfy triangle inequality. Sorry for the confusion and we will make it clear in the revision.
>
> - ### Regarding interpretation of embeddings and visualization:
> One idea for the interpretation of the non-Euclidean distances is that the dissimilarities are metric distances between two sets of points. One way to view this is that each data element is actually a distribution with some uncertainty. The non-Euclidean distance actually penalizes such uncertainty. This is our ongoing work, which will likely to be a separate follow up paper.
>
> Inspired by the referenced work, we also plot some 2-dimensional neudMDS embeddings of word embeddings from a Bert model trained for some text classification tasks. The plots include positive-positive, positive-negative, and negative-negative eigenvectors of the neucMDS. See the attached file in general response.

---

> > ### Comment · Reviewer_EWdH · 2024-08-12
> >
> > Thank you for the response.
> > I'm glad that the provided reference was helpful.

---

> > > ### Author Response · Authors · 2024-08-13
> > >
> > > Thank you for the comment and suggestions. We appreciate it!

---

### Official Review · Reviewer_RtpY · 2024-07-12

**Soundness:** 3
**Presentation:** 3
**Contribution:** 3
**Rating:** 6
**Confidence:** 4

**Summary:**

The paper introduces Neuc-MDS, a novel extension of classical Multidimensional Scaling (MDS) designed to handle non-Euclidean and non-metric dissimilarities in datasets. The goal is to create accurate low-dimensional embeddings while minimizing the STRESS (sum of squared pairwise error)

**Strengths:**

1.	Neuc-MDS extends the concept of the inner product to a broader class of symmetric bilinear forms, allowing the incorporation of both positive and negative eigenvalues from the dissimilarity Gram matrix. This generalization helps capture the underlying structure of non-Euclidean data more effectively than classical MDS, which typically discards negative eigenvalues.
2.	Neuc-MDS is specifically designed to handle non-Euclidean and non-metric dissimilarities, making it versatile for a wide range of applications where traditional MDS falls short. The method is backed by a thorough theoretical analysis, providing guarantees for minimizing STRESS. This includes a detailed decomposition of the STRESS error and the demonstration of optimality in eigenvalue selection.
3.	Neuc-MDS is capable of working with a variety of dissimilarity measures that are commonly used in practice but are not Euclidean, such as cosine similarity, Hamming distance, and Jaccard index. This broadens its applicability to different fields and types of data.

**Weaknesses:**

1.	The theoretical guarantees provided by Neuc-MDS are based on certain assumptions about the data and dissimilarity matrices. If these conditions are not met in practice, the performance and guarantees may not hold.
2.	Some theoretical results rely on properties of random matrices (e.g., Wigner's Semicircle Law). The applicability of these results to structured or real-world datasets, which may not exhibit such random properties, is unclear. The use of Lorenzian distance or other non-standard measures may not satisfy traditional distance properties (e.g., triangle inequality), potentially leading to confusion or misinterpretation in applications that rely on classical distance metrics.
3.	The algorithm involves eigenvalue decomposition and optimization over eigenvalue selections, which can be computationally intensive for large datasets. The scalability of Neuc-MDS, especially for very large datasets, may pose a practical challenge.
4.	The implementation of Neuc-MDS involves several complex steps, including eigenvalue selection and optimization. This complexity may hinder its adoption by practitioners who require straightforward and easily implementable solutions.

**Questions:**

1.	How does Neuc-MDS perform on extremely large datasets compared to cMDS and other dimensionality reduction methods in terms of runtime and memory usage? Are there specific strategies or heuristics recommended for choosing the subset of eigenvalues in very large-scale applications to balance between computational efficiency and accuracy?
2.	How do the negative eigenvalues impact the interpretability of the resulting low-dimensional embeddings in practical applications, such as in visualization or clustering? What are the recommended practices for handling and interpreting dissimilarity matrices with negative entries generated by Neuc-MDS in downstream tasks?
3.	To what extent do the theoretical guarantees of Neuc-MDS hold for real-world datasets that do not conform to the random matrix assumptions used in the analysis?
4.	How does Neuc-MDS compare with other non-linear dimensionality reduction methods (e.g., t-SNE, UMAP) in preserving the global and local structure of non-Euclidean datasets?

**Limitations:**

See weakness

---

> ### Author Rebuttal · Authors · 2024-08-07
>
> We sincerely thank you for your careful reading and valuable feedback. In the following we address the concerns.
>
> - ### Regarding Computation and Scalability
> We discuss the computational complexity and provide more ideas (e.g. using ideas similar to landmark MDS) on reducing the runtime for very large datasets in the general response (1). Basically, these concerns have been addressed by previous work.
>
> - ### Regarding complexity of implementing Neuc-MDS
> Our algorithm, compared to MDS, differs only in the way that eigenvalues are chosen. This algorithm is a simple greedy algorithm (Algorithm 2), that is both easy to understand and to implement, with linear running time in the number of eigenvalues to choose. We respectfully disagree that this is complex and thus not practical for implementation. We have also shared our code through the link in the paper for anyone who'd like to try it out.
>
> - ### Regarding comparison with other methods
> We compare with two non-linear methods: Smacof and t-SNE and provide the results in the general response (3). Neuc-MDS still outperforms those non-linear methods on non-Euclidean datasets, which implies that the issue raised from the non-Euclidean properties of the underlying spaces of datasets cannot be easily solved by non-linearity.
>
> - ### Regarding Interpretation of dissimilarities
> One idea for the interpretation of the non-Euclidean distances is that the dissimilarities are metric distances between two sets of points. One way to view this is that each data element is actually a distribution with some uncertainty. The non-Euclidean distance actually penalizes such uncertainty. This is our ongoing work, which will likely to be a separate follow up paper.
>
> - ### Regarding assumptions on dissimilarity matrices
> We remark that our assumptions on the input dissimilarity matrices are very general, especially compared to the Euclidean distance assumption in classical MDS and a lot of machine learning settings. That gives our methods potential to be applied to more general situations like non-Euclidean geometry or graphs. Also, our algorithm for choosing eigenvalues minimizes the error terms C1+C2, which is always a lower bound of STRESS=C1+C2+C3 (since C3 is non-negative). This does not depend on any assumption on the input data.
>
> - ### Regarding Lorenzian distances and non-metric dissimilarities
> Indeed many machine learning applications use metric distances or even only Euclidean distances, thus making it an (arguably) `comfort zone'. At the same time, many dissimilarity measures including Minkowski distance (Lp), cosine similarity, Hamming, Jaccard, Mahalanobis, Chebyshev, and KL-divergence are not Euclidean and sometimes even not a metric. Real world examples of genome data set [17] used in the experiments are also non-Euclidean. When such dissimilarities have their respective applications and are indeed used in practice (possibly more than anticipated), it is necessary to study such dissimilarities under dimension reduction. It is common if we look at the scientific history where scientists step outside comfort zone and ask -- what is beyond this assumption?
> For the same reason, we have different views regarding that the use of non-metric distances potentially leading to confusion or misinterpretation. We hold an optimistic view that broadening our study of dissimilarity beyond Euclidean distances can bring us new opportunities and our work can be helpful in this direction.
>
> [17] B. C. Feltes, et al., CuMiDa: an extensively curated microarray database for benchmarking and testing of machine learning approaches in cancer research. Journal of Computational Biology, 26(4):376–386, 2019.
>
> - ### Regarding Random matrices
> We fully agree that real world data is unlikely to generate a fully random matrix. In addition, we would like to add two interesting implications of this theoretical study. First, Euclidean distances support aggressive dimension reduction as evidenced by the Johnson Lindenstrauss Lemma (from $n$ dimensional space to $O(\frac{1}{\varepsilon^2}\log n)$ dimensional space with $1+\varepsilon$ distortion). The analysis for a random symmetric matrix points out that aggressive dimension reduction is indeed a luxury for Euclidean or other structured data, even if we use inner products that are not limited to Euclidean distances. Second, any real world data carries some measurement noise. When the scale of such random noise becomes non-negligible, STRESS error introduced by such noise cannot be small with aggressive dimension reduction. We would recommend practitioners to examine the spectrum of eigenvalues to gain insights on the power or limit in reducing dimensions.

---

> > ### Comment · Reviewer_RtpY · 2024-08-10
> >
> > Thanks for the clarification, I'll keep my score.

---

> > > ### Author Response · Authors · 2024-08-13
> > >
> > > Thank you for the comments! We are available to address are any additional questions.

---

### Official Review · Reviewer_3cJU · 2024-07-14

**Soundness:** 2
**Presentation:** 2
**Contribution:** 2
**Rating:** 4
**Confidence:** 4

**Summary:**

The authors introduce Non-Euclidean-MDS (Neuc-MDS), an extension of Multidimensional Scaling (MDS) that accommodates non-Euclidean and non-metric outputs, efficiently optimizes the choice of (both positive and negative) eigenvalues of the dissimilarity Gram matrix to reduce STRESS. The results seem to be promising and the error analysis looks solid.

**Strengths:**

1. the error analysis seems solid.
2. the topic is important which can extend to non-Euclidean MDS
3. the experiments look promising

**Weaknesses:**

1. The overall writing is not easy to follow, for example from line 119-131, your statement is long, but I am still confused how to construct A. Apparently if A is Identity matrix, it degenerates into traditional one. Your contribution is to say A doesn't need to be Identity or even PSD, but how to construct A remains unknown to me.
2. I don't agree that A can be non-PSD, in which case the inner product of v with itself can be negative, I feel wired.
3. Line 137 you claim X can be recovered from B, however this is not precise.
4. the formatting can be improved, for example the fontsize Table 3 can be reduced and figure 3 can be better.

**Questions:**

What is the advantage of Neuc-MDS over Neuc-MDS+ or the opposite direction? From the experiments, I didn't find difference.

---

> ### Author Rebuttal · Authors · 2024-08-07
>
> We sincerely thank you for your careful reading and valuable feedback. In the following we address the concerns.
>
> - ### Regarding the problem definition (Definition 3.1) and description
> For a generic bilinear form $f_A(u, v)=u^T A v$, consider the induced dissimilarity $D_A(p_i,p_j):=f_A(p_i-p_j, p_i-p_j)$ between two points $p_i,p_j$ in $\mathbb{R}^n$
> (there was a typo in Definition 3.1 regarding the notation $\hat{D}_{i,j}$. we'll correct it in the revision).
>
> When $A$ is the identity matrix, the bilinear form gives a classical inner product and the dissimilarity is the (squared) Euclidean distance.
> When $A$ is a PSD, the vector space with this bilinear form can be mapped isometrically to an Euclidean space through the eigen-decomposition of $A$.
> In general, our paper focuses on looking for a bilinear form $f_{\hat{A}}$ defined by a low-rank $\hat{A}$ to approximate the bilinear form $f_{A}$ (or its induced dissimilarity $D_A$) of the underlying space of the given data.
>
> How to construct the low-rank $\hat{A}$ or dissimilarity matrix $\hat{D}$ is the main task of this paper (Algorithm 1). We construct $\hat{A}$ together with embeddings implicitly in our $\hat{D}$. To see that, based on the relation between dissimilarities and bilinear forms (gram matrices), we have $\hat{X}^T\hat{A}\hat{X}=-C\hat{D}C/2=U{diag({\lambda}\odot {w})}U^T$.
> The last equality is given by the eigen-decomposition with $U$ being the matrix of real eigenvectors, and ${w}$ indicating the selection of our algorithm on the eigenvalues ${\lambda}$. Then for $\hat{X}=\sqrt{diag({|\lambda|}\odot {w})}U^T$, the bilinear form $\hat{A}=diag(sign({\lambda}))$.
>
> - ### Regarding non-PSD A
> One of the main motivations of this work is based on some initial observations of importance of studying non-PSD bilinear forms which appeared in several research domains, and we believe it will benefit the machine learning community in general. We give more discussion in the general response (2).
>
> - ### Regarding Line 137
> Thank you for pointing out and yes, in general, we would say one instance of X (under some isometric transformations) realizing the distance matrix can be recovered by the algorithm (or more precisely, under some generic assumption, X is unique up to UXP where U is an orthogonal matrix, P is the matrix corresponding to the orthogonal projection from $\mathbb{R}^n$ to the subspace $\{x=(a_1, ...., a_n)\mid \sum_i a_i =0\}$). We would clarify it in the revision.
>
> - ### Regarding NeucMDS vs NeucMDS+
> Theoretically, both Neuc-MDS and Neuc-MDS+ look for a diagonal matrix of $\lambda'$ with at most $k$ non-zero entries to reconstruct a low rank dissimilarity matrix $D'$. Neuc-MDS uses $\lambda'$ by choosing at most $k$ non-zero eigenvalues from the input Gram matrix, while Neuc-MDS+ allows $\lambda'$ with arbitrary $k$ non-zero values -- or a general rank-$k$ linear transformation of the eigenvalues of the input Gram matrix. Thus Neuc-MDS+ looks for the optimal value in a larger domain.
> Empirically, we realize that both Neuc-MDS and Neuc-MDS+ give smaller STRESS values but the number of negative distances using Neuc-MDS+ is significantly fewer. Due to space limit the experiments are in Table 5 of Appendix E.3.
>
> - ### Regarding formatting
> We appreciate your comments on formatting and will address these issues in the revised version.

---

> > ### Author Response · Authors · 2024-08-13
> >
> > We hope the response helped to clarity the issues. We are available to address any additional questions. Thanks for the efforts and suggestions in the review.

---

> ### Author Response · Authors · 2024-08-14
>
> Dear Reviewer,
>
> Thank you again for your constructive review. We appreciate your recognition of our error analysis and the importance of our topic.
>
> We hope our detailed response has addressed your concerns regarding the construction and the non-PSD properties of the bilinear form matrix A, the comparison between Neuc-MDS and Neuc-MDS+, and the noted formatting issues. We will include these updates in the revision. Please also see our general response, where we summarize the key steps taken to address the concerns of all the reviewers.
>
> As the deadline is approaching, if you feel that we have sufficiently addressed your concerns, we would greatly appreciate if you could update your score.
>
> Best regards.

---

### Official Review · Reviewer_wzYh · 2024-07-25

**Soundness:** 4
**Presentation:** 3
**Contribution:** 3
**Rating:** 7
**Confidence:** 5

**Summary:**

This paper introduces Non-Euclidean Multidimensional Scaling (Neuc-MDS), an extension of classical Multidimensional Scaling (MDS) that can handle non-Euclidean and non-metric data. The key ideas and contributions are:

1. It generalizes the inner product to more general symmetric bilinear forms, allowing the use of both positive and negative eigenvalues of dissimilarity matrices.

2. Neuc-MDS efficiently optimizes the choice of eigenvalues to minimize STRESS (sum of squared pairwise errors). The authors set up the problem as a quadratic integer program but show that it has an optimal solution and that it can be found in a greedy fashion.

3. The authors provide theoretical analysis of the error and prove optimality in minimizing lower bounds of STRESS.

4. They introduce two algorithms: Neuc-MDS and an advanced version Neuc-MDS+.

5. Theoretical analysis is provided for the asymptotic behavior of classical MDS and Neuc-MDS on random symmetric matrices.

6. Experimental results on 10 diverse datasets show that Neuc-MDS and Neuc-MDS+ outperform previous methods on STRESS and average distortion metrics.

7. The proposed methods resolve the "dimensionality paradox" issue of classical MDS, where increasing dimensions can lead to worse performance.

8. The approach is applicable to both image and text data, and can handle various non-Euclidean dissimilarity measures.

9. The paper provides detailed proofs, algorithm descriptions, and experimental results in the appendices.

Overall, this work extends MDS to non-Euclidean spaces, providing both theoretical guarantees and practical improvements over existing methods for dimension reduction and data embedding tasks.

**Strengths:**

This paper is well-written and it clearly outlines the problem and the solution. The analysis is non-trivial and the experiments are thorough.

**Weaknesses:**

While the paper presents a novel approach with several strengths, there are a few potential weaknesses or limitations that can be identified:

1. Computational complexity: For large datasets, the method may still be computationally intensive, as it requires eigendecomposition of the full dissimilarity matrix. While the authors mention the possibility of using approximation algorithms for partial SVD, they don't provide detailed analysis or experiments on very large-scale datasets.

2. Limited comparison with non-linear methods: The paper primarily compares Neuc-MDS with classical MDS and other linear dimension reduction techniques. A comparison with popular non-linear methods like t-SNE or UMAP could provide more context on its performance relative to the state-of-the-art in dimension reduction.

3. Interpretability: The use of general bilinear forms, while mathematically elegant, may make the resulting embeddings less interpretable compared to Euclidean embeddings, especially for domain experts not familiar with non-Euclidean geometries.

4. Downstream tasks: The paper focuses on the quality of the embedding itself (via STRESS and distortion metrics) but doesn't extensively explore how these embeddings perform in downstream machine learning tasks compared to other methods.

5. Negative distances: The method can produce negative distances, which may be problematic for some applications or require special handling in downstream tasks. While this is mentioned, strategies for dealing with negative distances are not fully explored nor is the impact on "real" data sets discussed.

6. A discussion of just how non-Euclidean "standard" data sets are would be useful. Is this problem of non-Euclidean distances a substantial one or not?

7. This brings me to a more detailed discussion of the experiments: I'd be interested in seeing a quasi-synthetic example of a real data set with non-Euclidean distances used (say, graph distances from a nearest neighbor graph). What impact does that make on the results?

**Questions:**

See above.

**Limitations:**

not applicable.

---

> ### Author Rebuttal · Authors · 2024-08-06
>
> Thank you for your valuable feedback, we respond to your comments as follows.
>
> - Computational Complexity
>
> Thanks for raising the concern. One-line argument is both Neuc-MDS and classical MDS run in $O(n^3)$ time due to eigen decomposition, therefore, no extra cost asymptotically. We also have more ideas on reducing the runtime, please refer to general response 1.
>
> - Additional Experiments with t-SNE
>
> In the submission we mainly focused on linear embeddings and compare with other MDS variants, to evaluate the impact of moving beyond Euclidean geometry. Some non-linear dimension reduction methods such as Isomap use MDS as intermediate steps. The submission already included another non-linear method: Smacof. We also run experiments with t-SNE on all datasets, and find that Neuc-MDS outperform others in terms of STRESS. Please refer the results in general response 3.
>
> - Interpretability
>
> Beyond the routine use of classical MDS for embedding and visual inspection of the input dataset, these dimension reduction methods also have many applications, simply to reduce the size of the representation needed. For such types of applications, Neuc-MDS can be a good alternative when the input dissimilarity is non-Euclidean.
>
> - Negative Distances
>
> We have discussions and provide two examples with references in the general response 2.
>
> - Downstream Tasks
>
> Exploring how non-Euclidean dimension reduction can benefit downstream applications is an important next step. We are very keen in exploring along this direction further, which is one of the important future directions.
>
> - Non-Euclidean Datasets
>
> In our simulation section, we tested on two generated datasets that carry non-Euclidean distances. The first one is a random distance matrix (called Random-Simplex), and the second one takes minimum distance between Euclidean balls (called Euclidean-ball). Details of the two datasets can be found in Appendix E.1. Xu, Wilson and Hancock [49] have suggested two sources of non-Euclidean non-metric distances, random noises and extended objects, which correspond precisely to these two datasets. The Euclidean ball dataset suggests a general source of non-Euclidean distances coming from distances of families of data points, where the distance of two sets $A$ and $B$ is defined as the Hausdorff distance $\min_{a\in A, b\in B} d(a, b)$. Such distances are not Euclidean and are not always a metric.
>
> In addition, the datasets we used from CuMiDa and image data set use dissimilarity values that are non-Euclidean, as can be seen from the negative eigenvalues in Figure 1 and Table 1. More discussion on this can be found in Appendix E.2.
>
> Please let us know if there is anything further we could help clarify. Thanks again for your time and efforts!

---

> ### Comment · Reviewer_wzYh · 2024-08-12
>
> Thank you. I'll keep my score. I do think that there are more directions in which to push this line of research and I don't want to ding the authors for not writing an extensive long paper (or three or four more) at this time!

---

> > ### Author Response · Authors · 2024-08-13
> >
> > Thank you for the comments and suggestions. We appreciate it!

---

### Author Rebuttal · Authors · 2024-08-06

We appreciate the efforts by reviewers and would like to clarify a few questions and answer to raised concerns. We are glad that this submission has been found to be:

- Well-written (Reviewer wzYh, EWdH)

- Showing solid/thorough experimental results (all reviewers)

- Accompanied with convincing theoretical analysis (Reviewer wzYh, 3cJU, RtpY)

- With codes are publicly available (Reviewer EWdH).

We would like to first respond to general concerns and then independent reviews. The common issues include (1) Computational complexity, (2) Practicality of the output distances and (3) Experiments on non-linear methods.

- ### Computational Complexity

Recall that Neuc-MDS has exactly the same running time as classical MDS, which is $O(n^3)$ due to eigen-decomposition, therefore, no extra cost asymptotically.

In addition, from Line 173-183, we included some suggestions of reducing running time by computing only the $k$ eigenvalues and eigenvectors needed.

On top of that, for extremely large datasets, we can use landmark version of non-Euclidean MDS. Landmark MDS [1] is a heuristic to speed up classical MDS. We choose $h$ landmarks and apply MDS on the landmarks first. The coordinates of the remaining points are obtained through a triangulation step with respect to the landmarks. We can use the same heuristic to speed up Neuc-MDS. We ran some additional tests with landmark Neuc-MDS on our random-simplex dataset and got the following results. If we use only 25\% points randomly chosen as landmarks the STRESS is only a factor of $1.0644$ of the STRESS obtained by Neuc-MDS. If we use only 10\% points as landmarks, the final STRESS is only $1.0898$ of the STRESS of Neuc-MDS. This shows that Neuc-MDS can also be significantly accelerated using the landmark idea, achieving nearly the same STRESS.

[1] Vin de Silva, et al. Sparse multidimensional scaling, using landmark points.

- ### Practicality of Output Distances

Squared dissimilarity matrices with negative entries do appear in a number of popular mathematical models. We present two examples here.

1. In elementary plane geometry,
the power of a point $P$ relative to a circle (center $O$, radius $r$) is $|PO|^2-r^2$, introduced by Jakob Steiner, reflects the relative distance of a given point from a given circle. It is positive outside the circle (squared tangential distance) and negative inside (negative squared distance to a point $S$ on the circle where $PSO$ forms a right triangle).
The power of a point leads to a natural structure power diagram -- Voronoi diagram of circles. It is a planar convex subdivision where inside the cell of a circle $c$, all points have power distance to $c$ to be the smallest compared to other circles. The power diagram is a fundamental notion in computational geometry with applications of data structures and algorithms for a collection of disks [2], that serve as foundation to many problems in computational biology --- molecules are often modeled as spheres. It is related to semi-discrete optimal transportation [3], fluid dynamics [5], capacitated clustering [4] and area-preserving mapping in machine learning [6], and early universe reconstruction.

[2] H. Imai, et al. Voronoi diagram in the Laguerre geometry and its applications.

[3] F. Aurenhammer, et al. Minkowski-Type Theorems and Least-Squares Clustering.

[4] C. Ni, et al. Capacitated Kinetic Clustering in Mobile Networks by Optimal Transportation Theory.

[5] B. L\'{e}vy,  Partial optimal transport for a constant-volume Lagrangian mesh with free boundaries.

[6] Na Lei, et al. A Geometric View of Optimal Transportation and Generative Model''.

2. Negative inner product norms have deep connections to the study of spacetime with special relativity. The Lorentzian $n$-space is an inner product space on $\mathbb{R}^n$ together with a non-PSD Lorentzian inner product norm: $|x|=-x^2_0 + x_1^2+\cdots + x^2_{n-1}$. The $n$-dimensional Lorentzian space is naturally classified into types based on the sign of the squared norm: spacelike (positive), timelike (negative) and lightlike (zero). The collection of all timelike vectors lie in the open subset formed by the interior of the light cone. For two similarly directed (both forward or backward in time) time-like vectors triangle inequality works in the opposite direction. Four-dimensional Lorentzian space is called the Minkowski space and forms the basis of special relativity. Furthermore, the collection of vectors in $\mathbb{R}^{n+1}$ with Lorentzian inner product of $-1$ has imaginary Lorentzian length and is precisely the hyperboloid model of the hyperbolic $n$-space $\mathbb{H}^n$ [7]. Along this line there are two potential directions of research.

First, with the increasing adoption of embedding in non-Euclidean spaces (e.g., hyperbolic spaces [8]) in machine learning models, we expect that non-Euclidean norms will find more applications. The second promising direction is in machine learning for physics model and data (AI4Sicence), see [9] for an example on the use of a feed-forward neural network for Petrov's classification of spacetimes.

[7] Ratcliffe, J. G. Foundations of Hyperbolic Manifolds.

[8] I. Chami, et al. Hyperbolic Graph Convolutional Neural Networks.

[9] Y. He, et al. Machine-Learning the Classification of Spacetimes.

We acknowledge that current machine learning practices predominantly take distances to be at least a metric space. The authors believe that it is necessary and the right time to expand our scope and consider non-Euclidean measures. This paper is making one step in this direction.

- ### Experiments on Non-linear Methods and visualization
We compare Neuc-MDS with t-SNE, a popular non-linear method. In the submission we compared with Smacof, also a non-linear method. Neuc-MDS outperforms other methods.
We also plot some 2-dimensional neudMDS embeddings of word-embeddings from a pretrained BERT model for a text classification task.  See results in the attached pdf.

---

### Decision · Program_Chairs · 2024-09-25

**Decision:**

Accept (poster)

**Comment:**

This paper proposes Non-Euclidean Multidimensional Scaling (Neuc-MDS) as an extension of classical Multidimensional Scaling (MDS), which can handle non-Euclidean and non-metric data. The idea is interesting and would be helpful. The weaknesses of the work are the computational complexity and the comparison with other non-linear methods.